# Predictors of perinatal mortality in the seven major hospitals of Lusaka Zambia: A case control study

**Musonda Makasa**[1,2,3]*, **Patrick Kaonga**[1], **Choolwe Jacobs**[1], **Mpundu Makasa**[4], **Bellington Vwalika**[3]

**1** Department of Epidemiology and Biostatistics, University of Zambia School of Public Health, Lusaka, Zambia, **2** University Teaching Hospital, Women and Newborn Hospital, Lusaka, Zambia, **3** Department of Obstetrics and Gynaecology, University of Zambia School of Medicine, Lusaka, Zambia, **4** Department of Community and Family Medicine, University of Zambia School of Public Health, Lusaka, Zambia

* mcmakasa@gmail.com

## Abstract

### Background

In 2023, approximately 2.3 million babies globally were lost before birth or within the first week of life, primarily due to preventable causes. Global perinatal mortality declined from 5.7 million in 2000 to 4.1 million by the end of 2015. However, despite this progress, for example 45% of all stillbirths were reported from high-income countries, which contribute less than 2% of the global burden of stillbirths. Perinatal mortality rates for sub-Saharan Africa and Zambia are at 37.4 and 33/1000 live births, respectively. The aim of this study was to determine the predictors of perinatal mortality at the seven major hospitals of Lusaka, Zambia.

### Methods

This was a multifacility unmatched case control study from September 2023 to January 2024. Cases included perinatal death (≥22 weeks gestation or ≥500g stillborn and death of neonate within seven days of life) and controls were live births. Stepwise multivariate logistic regression analysis with Stata version 14 determined predictors using adjusted odd ratios and p-values.

### Results

This study included 630 participants, with 210 cases and 420 controls, analysed in a 1:2 ratio. Antenatal care booking after 12 weeks gestation had almost three times odds of experiencing perinatal mortality (AOR 2.91, 95% CI: 1.97-4.29), p < 0.001) compared to early booking (<12 weeks). Walking as a means of reaching the healthcare facility had over three times the odds of perinatal mortality (AOR3.48, 95% CI: 1.87-6.49, p < 0.012) compared to using personal transport. Anaemia in pregnancy

**Data availability statement:** There are ethical restrictions which prevent the public sharing of minimal data for this study, because the data contains potentially identifiable patient information. Data are available upon request from the University of Zambia School of Public Health representative, Joseph Mumba Zulu, Professor of Community Health, via email (josephmumbazulu@gmail.com) for researchers who meet the criteria for access to confidential data.

**Funding:** The authors received no specific funding for this work.

**Competing interests:** The authors have declared that no competing interests exist.

carried a three-and-a-half-fold increased the risk of perinatal death (AOR 3.58, 95% CI: 1.72-7.44, p < 0.001) compared to mothers without anaemia. History of previous pregnancy loss was associated with a five-fold increased risk of perinatal death (AOR 5.05, 95% CI: 2.99-8.51, p < 0.001) compared to those without such a history.

## Conclusion

This study revealed that late antenatal care booking, walking as means of transport to access health facility, anaemia in pregnancy, and previous history of loss of baby before birth perinatal death were the main predictors with statistical significance of perinatal death experience. The study findings highlight the need for policies that promote early antenatal care, prevent anaemia in pregnancy, improve transport access to hospitals, and further research into context-specific barriers and effective interventions.

## Introduction

Perinatal mortality remains a major global health challenge, particularly in low-income settings, and imposes significant direct, indirect and intangible economic burdens on women, their families, and the wider society [1,2]. In 2019, approximately 5,400 stillbirths occurred daily, amounting to an estimated 2 million annually [3,4]. In 2023 alone, an estimated 2.3 million newborns died globally [5,6]. Despite efforts aligned with the United Nations (UN) Sustainable Development Goals (SDGs), particularly SDG 3.2, which aims to end preventable newborn deaths by 2030, progress in sub-Saharan Africa (SSA) has been limited [7]. Perinatal mortality refers to the loss of a foetus after 22 weeks of gestation, or with a birthweight ≥ 500g; as well as the death of a newborn within the first 7 days of life. Perinatal mortality comprises of stillbirths and early neonatal deaths. Early neonatal mortality specifically refers to the death of a newborn within the first 7 days of life [8,9]. To facilitate international comparability, the International Classification of Diseases, 11th Revision (ICD-11) recommends defining stillbirths as foetal deaths occurring at ≥28 completed weeks of gestation, or - where gestational age is unknown – a birthweight of ≥1000g [9,10]. This standardized threshold ensures consistency in reporting foetal deaths globally. Stillbirths are typically classified into two categories: fresh and macerated. A fresh stillbirth (FSB) shows no signs of maceration (skin changes) at the time of delivery and is presumed to have occurred recently (in less than 8 hours), typically intrapartum (during labour). In contrast, a macerated stillbirth (MSB) exhibits signs of maceration - such as skin discolouration, peeling, and soft tissue breakdown – an indication that death occurred more than 6–8 hours prior to delivery [11–13].

Although global perinatal mortality declined from 5.7 million in 2000 to 4.1 million by 2015, this progress has been uneven [14]. Similarly, the global stillbirth rate also declined from 24.7 per 1000 live births in 2000 to 18.4 in 2015 [7]. In 2000, the WHO estimated global perinatal mortality rate (PMR) of 47 per 1000 live births, with a

significantly higher rate of 62 per 1000 in Africa [8]. A more recent systematic review and meta-analysis by Tiruneh, Assefa [15] reported observed and adjusted PMR of 58.4 and 42.9 per 1000 live births respectively across SSA. These figures demonstrate and emphasize the persistent burden of perinatal mortality in resource-limited settings and highlight the critical need for continued investment in maternal and newborn health research and interventions.

One of the key contributing factors to the substantial disparity in PMR between high-income countries (HIC) and low- and middle-income countries (LMIC) is the historical neglect of stillbirths in global health agendas. Until recently, stillbirths were not included in the MDGs and were also omitted from major global health tracking systems, such as those maintained by either the UN bodies or the Global Burden of Disease (GBD) [16]. The lack of stillbirths' recognition contributed to their invisibility in global health priorities until very recently (2014), when the Every Newborn Action Plan (ENAP) made a critical shift to include a global target to reduce stillbirths to 12 or fewer per 1000 live births in every country by 2030. Additionally, ENAP called for countries to reach a Stillbirth Rate (SBR) of 14 per 1000 live births by 2020 as an interim milestone [17]. This delayed recognition of stillbirths as a significant global health challenge, combined with limited social recognition, investment, and programmatic action, further exacerbated the issue [7]. Additionally, in 2017, the Maternal Mortality Estimation Interagency Group (MMEIG) reported 295,000 maternal deaths worldwide, with approximately 196,000 (66.4%) occurring in SSA [18]. In low-income settings, over a third of these maternal deaths and half of the stillbirths occur before or during childbirth [13,19]. Moreover, nearly two thirds of the causes of maternal death also contribute to perinatal deaths [20].

While the global perinatal mortality has declined substantially in HIC - for example, to 3.4 per 1000 live births in Hong Kong and 5.5 per 1000 in the United States [21,22] – these countries account for only approximately 2% of global perinatal mortality cases [23]. In stark contrast, SSA continues to bear the highest PMR globally, with an average of 37.3 per 1000 live births. Cote D'Ivoire reports the highest figure at 68.1 per 1000 live births, as documented in a recent systematic review and meta-analysis [24]. Within Southern Africa, the regional the PMR stands at approximately 30.3 per 1000 live births [15,25].

Zambia continues to face a high PMR, estimated at 33 per 1000 live births [26], which exceeds regional averages and falls significantly short of short of the target set by the SDGs and Zambia Vision 2030 [27]. This study aimed to investigate the predictors of perinatal death across all the major hospitals of Lusaka, Zambia. Specifically, it sought to identify the social and demographic factors, as well as maternal and foetal characteristics, that are associated with perinatal deaths in this setting. Lusaka is Zambia's most populous district and the one with the highest birth rate. In addition, the research sought to generate evidence to inform targeted programmatic interventions and health systems strategies aimed at reducing the PMR within this high-burden setting.

## Materials and methods

This was a multifacility study conducted at the seven major hospitals in Lusaka urban district, Zambia. Data collection commenced on 9 September in 2023 and concluded on 31 January 2024. Participants (mothers with perinatal deaths) were prospectively recruited as cases at the time the perinatal deaths occurred. Controls were selected as the next mothers who delivered live births at the same facility after each case was recruited. Eligible participants who were agreeable to participate after being provided with detailed information via a participant information sheet were then recruited after obtaining informed written consent. Data was collected using an interviewer-structured questionnaire. The hospitals involved all the major hospitals of Lusaka which include: two tertiary facilities - Women and Newborn and Levy Mwanawasa University Teaching Hospitals - and five first level hospitals: Chawama, Chilenje, Chipata, Kanyama, and Matero first level hospitals. According to the Census Report, 2022 the Lusaka is the most populous city in Zambia, with a female population of 1,590,922 [28].

### Study design

This was an unmatched prospective case-control study with a 1:2 ratio, aimed at determining the predictors of perinatal mortality at the selected study site. Cases were defined as stillbirth after 24 weeks of gestation; ≥ 500g where gestational

age was not clear; and early neonatal deaths within seven days of life from the study sites. For each case, two controls were selected through systematic random sampling, involving women who had live births immediately before or after occurrence of a case. Controls were recruited from the same facility where the cases occurred, and for early neonatal deaths, controls were mothers who delivered within 24hrs of case's birth.

A midwife from each labour and postnatal ward at each facility was trained and enrolled as data a collector. Data collectors provided participants with an information sheet and obtained informed consent. In rare cases where translation was necessary, data collectors translated the information into the appropriate language that participants understood and were more conversant with. Data collection for postnatal mothers occurred whenever a case was identified (within the early neonatal period) and controls were subsequently recruited. In rare cases, particularly following an early neonatal death, controls were followed up before discharge or during clinic visits. These controls were mothers with live babies who delivered within 24 hours of a recruited case, ensuring their recruitment occurred around the same time as the case. Data were collected by reviewing files followed by interviewer administered collection of data using a standardized structured questionnaire on a google sheet that was managed by the Principal Investigator (PI). The PI monitored the Google data collection sheet daily.

## Study population

Study population comprised women aged 18 years and older who sought and received maternal and/or childbirth services at the study sites and met the inclusion criteria. Refer to the following: Fig 1 flowchart illustrates the source of the sample, eligibility screening process and how the final sample was arrived at.

## Inclusion criteria

Cases included stillbirths from mothers aged above 18 years, with a gestational age above 24 weeks. In the absence of gestational age, a birthweight of ≥500g was considered eligible for this study.

## Exclusion criteria

Women were excluded if they withdrew from the study, were under the age of 18, had a pregnancy of less than 22 weeks of gestation, or in the absence of gestational age, had a birthweight of less than 500g. Women were excluded if they were unavailable during the study period, were unable to participate due to ill health, or were too emotionally distressed to provide reliable information, even after reviewing the participant information sheet. Additionally, women who did not consent to participate were excluded.

## Sample size calculation

The sample size for the study was calculated based on the assumptions that two controls per case (r=2), with 80% power of the study (type II error 20%, i.e., β=0.2), confidence interval of 95% with 5% type I error (i.e., α=0.05). With an assumed odds ratio of 2.5 for differences based on findings in previous similar studies. They reported for example associated potential risk factors included extremes of birthweight, post term delivery, infections, haemorrhage and previous history or perinatal mortality [29,30]. And using Zambia's PMR of 33 per 1000 live births [31], the following formula by Charan and Biswas [32] was used:

$$n = \frac{r+1}{r} \left[ \frac{p^* \left(1 - p^*\right) \left(Z_\beta + Z_{\frac{\alpha}{2}}\right)^2}{\left(P_1 - P_2\right)^2} \right]$$

where $n$=is the sample size, $p^*$average proportion exposed which was given by the sum of proportion exposed cases ($P_1$) and proportion of control $P_2$ divided by 2 (i.e., = $\frac{P_1 + P_2}{2}$), $Z_\beta$ is the standard normal value for power (for power of 80%,

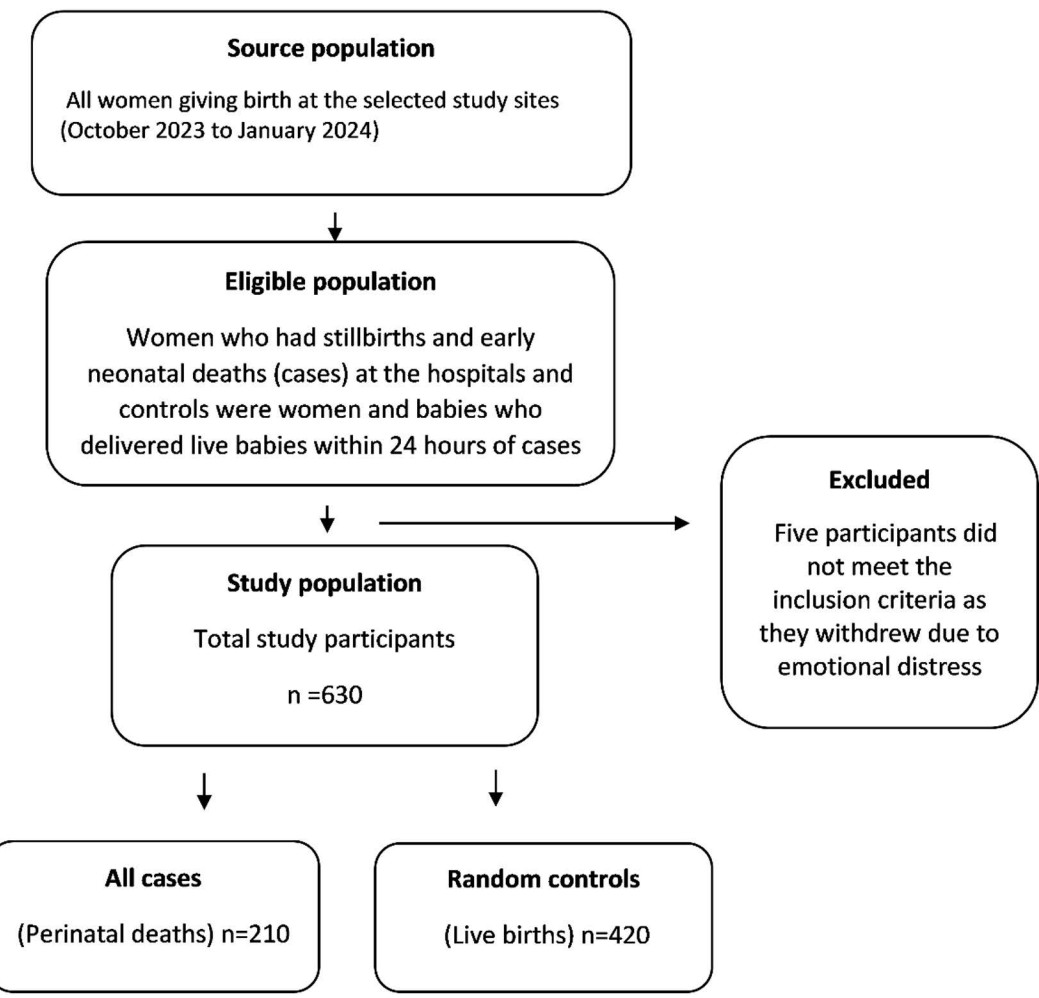

**Fig 1. Flowchart illustrating the source of the sample.**

$Z_\beta = 0.84$), $Z_{\alpha/2}$ is the standard normal value for the level of significance which is usually given as $\alpha = 0.05$ and its value is 1.96, and r is the ratio of controls to cases, in our case the ratio is 2 to 1 which means our r = 2; To find $P_2$, we use the formula:

$P_2 = \frac{OR \times P_1}{P_1(OR-1) + 1}$ where OR is the odds ratio and $P_1$ is the proportion with a specific risk factor in the control participants. Assuming OR = 3.2 based on pooled analysis with strong association of perinatal mortality due to lack of antenatal care [33]: $P_1 = 0.033$ in the formula above we get the value of $P_2$ as 0.09845

$$P^* = \frac{0.033 + 0.09845}{2} = \frac{0.13145}{2} = 0.0657$$

Replacing the above information in the prescribed formula above the following was the targeted sample size:

$$n = \frac{2+1}{2} \left[ \frac{0.0657(1-0.0657)(0.84+1.96)^2}{(0.09845-0.033)^2} \right]$$

$$n = 168.4874$$

$$n = 169$$

Therefore, the calculated sample size was 169 cases and 338 using the 1:2 ratio, which gave 507. After factoring in 10% additional in case of participant fall out, the total sample size calculated was 558.

## Sampling technique

To determine the sample size for each facility, using Probability Proportional to Size (PPS) sampling, we accounted for the delivery volume at each facility to ensure that larger facilities, which manage more deliveries, had a proportionally higher chance of selection. Hence, the following was applied in our study:

1. **To establishing facility delivery size**: We first reviewed delivery records from each facility covering January to December 2023 to estimate the delivery size at each one. This information served as the basis for determining the relative size of each facility. The allocation of cases across the seven participating health facilities was based each facility's share of the total perinatal deaths recorded across all sites. Specifically, the number of perinatal deaths at each facility was divided by the total number of perinatal deaths across all facilities (n = 1271) to obtained the proportion of the total the total burden attributable to each facility. The proportion was then multiplied by the total number of target cases (n = 210) to determine the number of cases to be recruited per site. The number of controls per facility was subsequently determined by applying a 1:2 case-to-control ratio.

For example, the Women and Newborn Hospital, which accounted for 363 of the 1271 perinatal deaths (28.6%), was allocated 59 cases and 118 controls, Similarly, Kanyama Hospital, contributed 18.3% of the total perinatal deaths (n=233), was allocated 39 cases and 78 controls.

2. **To ensure proportional allocation**: This proportional distribution ensured facilities with higher delivery numbers were assigned a greater probability of selection compared to those with fewer deliveries. This allocation ensured that our sample would reflect the distribution of delivery cases across facilities, avoiding under-representation from larger facilities. The following Table 1 shows the PPS distribution of the 630 participants – based on the proportion of total perinatal deaths and deliveries – across the seven facilities.

3. **To sample participants within facilities**: Once the facilities were selected with PPS, we then identified participants within each facility for inclusion in the study. This selection process within facilities allowed us to capture representative samples proportionate to the overall patient volume, aligning with the facility's estimated annual deliveries.

By using PPS sampling, we ensured a sample distribution that mirrors the scale of service utilization across facilities, thereby enhancing the representativeness and reliability of our findings.

A total of 630 participants (comprising 210 and 420 controls) were targeted for inclusion and subsequently enrolled across all study sites. Data collection was through an interviewer administered questionnaire using a password controlled google sheet. Due to rarity of the outcome, purposeful sampling was used for the cases that met the inclusion criteria. For each case enrolled, two controls were randomly selected within the same 24-hour shift in which the case occurred.

## Study variables

The outcome variables include macerated (a stillbirth with features of maceration), fresh stillbirths (without features of maceration), and early neonatal death (occurred within 7 days of life). Distal level variables included age, marital status, parity, gravida, weight, BMI, employment status, educational level, distance of place of residence to health facility, and mode of transport. Intermediate level variables were antenatal booking data, number of antenatal contacts, history

**Table 1. Proportional to size sampling based on total deliveries and perinatal deaths statistics from January – December 2023.**

| Facility | Total Deliveries | Proportion of Deliveries | Perinatal Deaths | Proportion of Perinatal Deaths | Allocated Cases (n = 210) | Allocated Controls (n = 420) | Total Sample (n = 630) |
|---|---|---|---|---|---|---|---|
| Chawama | 7,590 | 0.1473 | 52 | 0.0409 | 9 | 18 | 27 |
| Chilenje | 5,059 | 0.0982 | 12 | 0.0094 | 2 | 4 | 6 |
| Chipata | 9,117 | 0.1769 | 169 | 0.1329 | 28 | 56 | 84 |
| Kanyama | 11,336 | 0.2201 | 233 | 0.1833 | 39 | 78 | 117 |
| Matero | 4,939 | 0.0959 | 192 | 0.1510 | 32 | 64 | 96 |
| Levy | 6,687 | 0.1298 | 250 | 0.1967 | 41 | 82 | 123 |
| WNH | 6,797 | 0.1319 | 363 | 0.2858 | 59 | 118 | 177 |
| **Total** | **51,525** | **1.0000** | **1,271** | **1.0000** | **210** | **420** | **630** |

of pregnancy loss beyond viability, history of abortion, malaria during pregnancy, Human Immunodeficiency Virus (HIV) status, tuberculosis during pregnancy, syphilis, alcohol intake during pregnancy, illicit drugs, smoking, folic and ferrous sulphate use during pregnancy, malaria treatment during pregnancy, and sickle cell anaemia. Rhesus factor, antepartum haemorrhage, preeclampsia, and eclampsia, preterm delivery, history of low birth weight, and inter-pregnancy interval were the rest. While proximal level variables included gestational age at the time of birth, outcome, Apgar score, time lag to Neonatal Intensive Care Unit (NICU), age of early neonatal at the time of demise, sex, gross congenital anomalies, congenital syphilis, prematurity, low birthweight, and cause of death.

## Data collection tool

A standard researcher administered questionnaire was set up using the google sheet platform with protected access to the selected data collectors only. The PI exclusively handled daily monitoring in real-time. After completion of data collection, the data set was downloaded to an excel format for screening and cleaning and then coded to construct a do file for execution in Stata version 14.

## Data analysis

Descriptive statistical analyses were conducted to summarize the proportions and frequencies of cases and controls. Continuous variables such as age, weight, parity, and gravidity were categorized based on clinical reasoning before being introduced into the model. A bivariate analysis using Chi-square test was performed to identify the factors associated with perinatal mortality. All factors found to be associated with perinatal mortality in the bivariate analysis (*p*-value <0.05) at 95% confidence interval (95% CI) were then included in multivariable logistic regression model to adjust for potential confounders. Sequential elimination of predictors was employed to assess the strength and significance of the variables associated with perinatal deaths, followed by the discrimination and calibration.

The ability of the model to correct risk was assessed using the Hosmer and Lemeshow goodness of fit test. A non-significant *p*-value ($p > 0.05$) from this test indicated a good fit. Model stability was further evaluated by the measurement of the discrimination between participants with and without perinatal deaths using the area under the receiver operating characteristic (ROC) curve, with an acceptable values et at 0.7 [34]. The study followed the Strengthening the Reporting of Observational Studies in Epidemiology (STROBE) guidelines [35].

## Handling of missing data

Missing data in this study arose primarily due to incomplete records and non-responses to specific questions, in some instance, in the data collection tools. As reflected in the Table 2, some variables – such as parity, educational level, mode of transport, antenatal care (ANC) booking (the process if registering and scheduling regular appointments with a

**Table 2. Summary of the maternal socio-demographic characteristics and pregnancy factors associated perinatal mortality.**

| Predictors | Cases (n) 210 (%) | Controls (n) 420 (%) | Total (n) 630 (%) | χ² p-value |
|---|---|---|---|---|
| **Distal level variables** | | | | |
| **Age** | | | | |
| ≤ 19 | 30 (14.3) | 60 (14.3) | 90 (14.3) | |
| 20-34 | 138 (65.7) | 263 (62.6) | 401 (63.7) | 0.466 |
| > 35 | 27 (12.9) | 70 (16.7) | 97 (15.4) | |
| Missing | 15 (7.1) | 27 (6.4) | 42 (6.7) | |
| **Marital status** | | | | |
| Not married | 50 (23.8) | 93 (22.1) | 143 (22.7) | 0.741 |
| Married | 160 (76.2) | 318 (75.7) | 478 (75.9) | |
| Missing | 0 (0.0) | 9 (2.1) | 9 (1.4) | |
| **Parity** | | | | |
| 1–4 | 170 (81.0) | 309 (73.6) | 487 (77.3) | 0.702 |
| > 5 | 36 (17.1) | 62 (14.8) | 81 (12.9) | |
| Missing | 4 (1.9) | 49 (11.7) | 62 (9.8) | |
| **Employment Status** | | | | |
| Not employed | 151 (71.9) | 294 (70.0) | 445 (72.4) | 0.610 |
| Employed | 54 (25.7) | 116 (27.6) | 170 (27.6) | |
| Missing | 5 (2.4) | 10 (2.4) | 15 (2.4) | |
| **Educational level** | | | | |
| Primary | 65 (31.0) | 122 (29.0) | 187 (29.7) | |
| Secondary | 121 (57.6) | 210 (50.0) | 331 (52.5) | **0.017*** |
| Tertiary | 20 (9.6) | 75 (17.9) | 95 (15.1) | |
| Missing | 4 (1.9) | 13 (3.1) | 17 (2.7) | |
| **Intermediate level variables** | | | | |
| **Mode of transport** | | | | |
| Foot | 50 (23.8) | 56 (13.3) | 106 (16.8) | **0.001*** |
| Public transport | 139 (66.2) | 276 (65.7) | 415 (65.9) | |
| Personal | 20 (9.5) | 78 (18.6) | 98 (15.6) | |
| Missing | 1 (0.5) | 10 (2.4) | 11 (1.7) | |
| **ANC Booking** | | | | |
| < 12 weeks | 100 (47.6) | 83 (19.8) | 183 (29.0) | |
| > 12 weeks | 106 (50.5) | 323 (76.9) | 430 (68.3) | **0.001*** |
| Missing | 4 (1.9) | 14 (3.3) | 17 (2.7) | |
| **History of Stillbirth** | | | | |
| Yes | 64 (30.5) | 32 (7.6) | 96 (15.2) | **0.001*** |
| No | 143 (68.1) | 377 (89.8) | 520 (82.5) | |
| Missing | 3 (1.4) | 11 (2.6) | 14 (2.2) | |
| **History of Abortion** | | | | |
| Yes | 10 (4.8) | 7 (1.7) | 17 (2.7) | **0.018*** |
| No | 181 (86.2) | 393 (93.6) | 574 (91.1) | |
| Missing | 19 (9.0) | 20 (4.8) | 39 (6.2) | |
| **Proximal level variables** | | | | |
| **Malaria during pregnancy** | | | | |
| Yes | 21 (10.0) | 26 (6.2) | 47 (7.5) | 0.080 |
| No | 181 (86.2) | 384 (91.4) | 565 (89.7) | |

*(Continued)*

**Table 2.** (Continued)

| Predictors | Cases (n) 210 (%) | Controls (n) 420 (%) | Total (n) 630 (%) | χ² p-value |
|---|---|---|---|---|
| Missing | 8 (3.8) | 10 (2.4) | 18 (2.9) | |
| **HIV Status** | | | | |
| Positive | 30 (14.3) | 59 (14.0) | 89 (14.1) | 0.898 |
| Negative | 173 (82.4) | 351 (83.6) | 524 (83.2) | |
| Missing | 7 (3.3) | 10 (2.4) | 17 (2.7) | |
| **Tuberculosis during pregnancy** | | | | |
| Yes | 2 (1.0) | 1 (0.2) | 3 (0.5) | 0.227 |
| No | 201 (95.7) | 397 (94.5) | 598 (94.9) | |
| Missing | 7 (3.3) | 22 (5.2) | 29 (4.6) | |
| **Alcohol** | | | | |
| Yes | 14 (6.7) | 42 (10.0) | 56 (8.9) | 0.155 |
| No | 193 (91.9) | 368 (87.6) | 561 (89.0) | |
| Missing | 3 (1.4) | 10 2.4) | 13 (2.1) | |
| **Smoking** | | | | |
| Yes | 4 (1.9) | 2 (0.5) | 6 (1.0) | 0.087 |
| No | 204 (97.1) | 405 (96.4) | 609 (96.7) | |
| Missing | 2 (1.0) | 13 (3.1) | 15 (2.4) | |
| **Folate and Ferrous sulphate supplementation** | | | | |
| Yes | 197 (93.8) | 396 (94.3) | 593 (94.1) | 0.307 |
| No | 10 (4.8) | 13 (3.1) | 23 (3.7) | |
| Missing | 3 (1.4) | 11 (2.6) | 14 (2.2) | |
| **Anaemia in pregnancy** | | | | |
| Yes | 13 (6.2) | 60 (14.3) | 73 (11.6) | **0.003*** |
| No | 190 (90.5) | 349 (83.1) | 539 (85.6) | |
| Missing | 7 (3.3) | 11 (2.6) | 18 (2.9) | |
| **Sickle cell anaemia** | | | | |
| Yes | 1 (0.5) | 8 (1.9) | 9 (1.0) | 0.151 |
| No | 205 (97.6) | 400 (95.2) | 605 (96.0) | |
| Missing | 4 (1.9) | 12 (2.9) | 19 (3.0) | |
| **Antepartum Haemorrhage** | | | | |
| Yes | 11 (5.2) | 43 (10.2) | 54 (8.6) | 0.058 |
| No | 176 (83.8) | 358 (85.2) | 534 (84.8) | |
| Missing | 23 (11.0) | 19 (4.5) | 42 (6.7) | |
| **Preeclampsia** | | | | |
| Yes | 20 (9.5) | 47 (11.2) | 67 (10.6) | 0.863 |
| No | 160 (76.2) | 358 (85.2) | 518 (82.2) | |
| Missing | 30 (14.3) | 15 (3.6) | 45 (7.1) | |
| **Hypertension** | | | | |
| Yes | 14 (6.7) | 16 (3.8) | 30 (4.8) | 0.101 |
| No | 187 (89.0) | 393 (93.6) | 580 (92.1) | |
| Missing | 9 (4.3) | 11 (2.6) | 20 (3.2) | |
| **Diabetes Mellitus** | | | | |
| Yes | 1 (0.5) | 2 (0.5) | 3 (0.5) | 0.994 |
| No | 204 (97.1) | 404 (96.2) | 608 (96.5) | |
| Missing | 5 (2.4) | 14 (3.3) | 19 (3.0) | |

healthcare provider – midwife or doctor – to receive prenatal care throughout pregnancy), and various other variables – had varying degrees of missingness across both cases and controls. These gaps were largely a result of participants not recalling certain details, omissions in medical records, and failure to complete parts of the questionnaire. While the missing data were relatively minimal for most variable, their occurrence was accounted for during analysis to ensure validity and transparency of the findings.

### Ethical considerations

Ethical approval for this study was obtained from the University of Zambia Biomedical Research Ethics Committee (UNZABREC), reference number 3712-2023. Prior to this, clearance was secured from the National Health Research Authority (NHRA), under reference number NHRA 000012/16/03/2023. All procedures adhered to the ethical guidelines for research involving human participants, including informed consent. Participants were thoroughly briefed about the study's purposed, procedures, potential risks, and benefits, and written informed consent obtained before enrolment.

Confidentiality and privacy were strictly maintained throughout the study. All personal identifying information was anonymized, and the data collected was stored on password-protected devices accessible only to the principal investigator (PI) and designated research team members. To ensure data security, encrypted backups were created, and physical access to data storage location was restricted. Participants were also informed of their right to withdraw from the study at any point without any consequences. The study adhered to national and international research ethics standards.

## Results

### Description of study participants

This study comprised a total of a total of 630 study respondents. The total perinatal deaths 210 (cases) and 420 with live births (controls) made the ratio of cases to controls 1:2. The following Fig 2 hereafter shows the perinatal deaths distribution of fresh, macerated, and early neonatal deaths - summed across all study sites.

Fig 2 is a summary illustrating the frequency distribution of the Fresh Stillbirths (FSB), Macerated Stillbirth (MSB), and Early Neonatal Death (ENND) for this study. The proportion of the perinatal outcomes was 118 (56%) for ENNDs, 56 (27%) MSBs, and 36 (17%), FSBs respectively.

### Participants' socio-demographic characteristics

From the social-demographic attributes (distal level variables) of the respondents sampled majority of the respondents were between the age of 20–34 years 401 (63.7%). 19 years old and under were 90 (14.3%) and 97 (15.4%) were 35 years of age and above. Mean age for the study population was 27.3 years (SD: 6.8). Of the respondents 478 (75.9%) were Married and 143 (22.7%) were not married. A total of 487 (75.9%) had a parity between 1–4, while 81 (12.9%) were grand multipara. Additionally, 445 (72.4%) were not employed and 170 (27.6%) employed. For education, most of the respondents had only attained secondary level of education 331 (52.5%), primary 187 (29.7%), and 95 (15.1%) for tertiary. Most of the participants used public transport 415 (65.9%) while only 98 (15.6%) had personal transport and the others 106 (16.8%) on foot.

### Results of predictors of perinatal mortality from bivariate analysis

Bivariate analysis using Chi-Square Test revealed several variables associated with perinatal mortality. The outcomes outlined in the table below revealed that the level of education of the mother had a significant association with perinatal mortality (p<0.017). The form of transport used when going to the hospital (on foot) also had association with perinatal mortality (p<0.001). ANC booking beyond 12 weeks of gestation was statistically significant with p<0.001. Women who had prior history of perinatal death also had a statistically significant p-value (p<0.001). Abortion and Anaemia during

PLOS Global Public Health

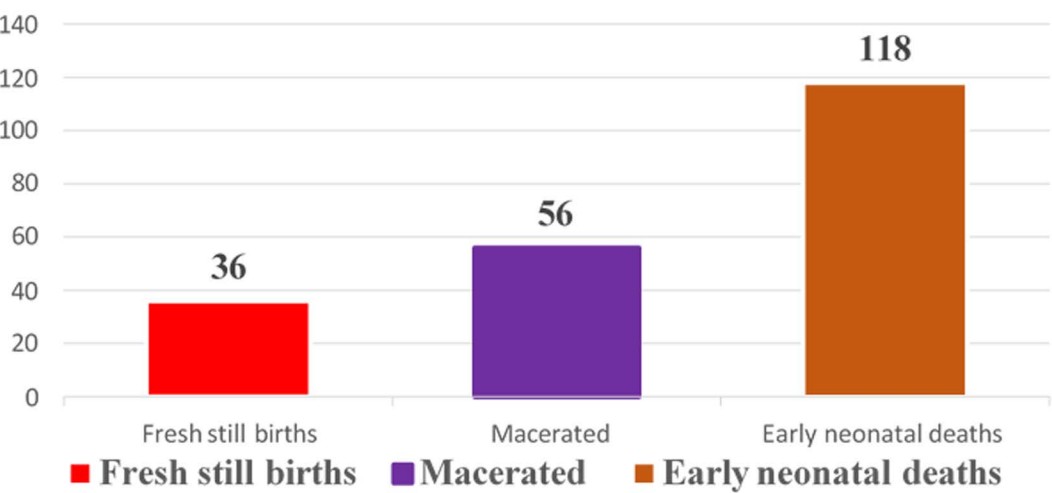

**Fig 2. Summary of distribution of Fresh, macerated stillbirths, and early neonatal deaths.**

pregnancy were found with p < 0.018 and p < 0.003 respectively. The rest of the independent variables had no statistical significance.

### Unadjusted logistic regression of variables that are predictors of perinatal mortality

According to the bivariate logistic regression for predictors of perinatal death (Table 3), respondents with a primary education level were almost two times likely to experience perinatal mortality compared to those in the tertiary level group (COR = 2.00, 95% CI: 1.12-3.56). The analysis also showed that the odds of experiencing perinatal death were over three times higher (COR = 3.48, 95% CI: 1.87-6.49) for women who walked (as means of transport) to access healthcare services and nearly two times as high (COR = 1.96, 95% CI: 1.15-3.34) for those that used public transport, compared to women who had personal transport.

Additionally, starting ANC after 12 weeks of gestation had over three and half odds (COR = 3.68, 95% CI: 2.56-5.30) of perinatal mortality compared to those that started earlier. Women with previous history of stillbirth were almost five times more likely to experience perinatal mortality (COR = 5.27, 95 CI: 3.31-8.402) compared to those without such a history. Women with history of abortion were also three times more likely (OR=3.10, 95% CI: 1.16-8.28) to experience perinatal mortality compared to those who had no history of abortion before. This study also found that women who experienced anaemia during pregnancy had more than two and half times higher odds of perinatal mortality (COR = 2.51, 95 CI: 1.34-4.69) compared to those without anaemia.

To understand the combined impact of predictors of perinatal mortality and to control for confounding, multivariate logistic regression analysis was conducted. Table 3 is the illustration of the final multivariate model that best describes predictors of perinatal mortality. Following a thorough analysis and correction for the impact of extraneous variables, four variables stood out as predictors of perinatal mortality. Educational level after multivariate logistic regression analysis did not show significance as a predictor of perinatal mortality. Mode of transport to the healthcare facility, on the other hand, showed significance as a predictor of perinatal mortality. Women who walked to the hospital had three times the odds of perinatal mortality compared to those who used personal transport (AOR 3.48, 95% CI: 1.87–6.49, p < 0.001). Antenatal care (ANC) booking after 12 weeks of gestation was significantly associated with an increased risk of perinatal mortality.

**Table 3. Multivariable logistic regression analysis for factors associated with perinatal mortality in the selected study sites.**

| Univariable logistic regression | | | | Multivariable logistic regression analysis | | |
|---|---|---|---|---|---|---|
| Predictor variable | Crude Odds Ratio (COR) | 95%CI | p-value | Adjusted Odds Ratio (AOR) | 95% CI | P-value |
| **Educational level** | | | | | | |
| Tertiary | Ref | | | | | |
| Primary | 2.00 | (1.12-3.56) | 0.019 | 1.66 | (0.83-3.31) | 0.152 |
| Secondary | 2.16 | (1.26-3.71) | 0.005 | 1.87 | (0.98-3.57) | 0.058 |
| **Mode of Transport** | | | | | | |
| Personal | Ref | | | | | |
| Foot | 3.48 | (1.87-6.49) | <0.001 | 2.75 | (1.23-5.39) | **0.012*** |
| Public | 1.96 | (1.15-3.34) | 0.013 | 1.70 | (0.92-3.16) | 0.091 |
| **ANC Booking** | | | | | | |
| < 12 weeks | Ref | | | Ref | | |
| > 12 weeks | 3.68 | (2.56-5.30) | <0.001* | 2.91 | (1.97-4.3) | **<0.001*** |
| **History of Stillbirth** | | | | | | |
| No | Ref | | | Ref | | |
| Yes | 5.27 | (3.31-8.40) | <0.001* | 5.05 | (2.99-8.51) | **<0.001*** |
| **History of abortion** | | | | | | |
| No | Ref | | | Ref | | |
| Yes | 3.10 | (1.16-8.28) | 0.024 | 0.86 | (0.24-3.06) | 0.814 |
| **Anaemia during pregnancy** | | | | | | |
| No | Ref | | | Ref | | |
| Yes | 2.51 | (1.34-4.69) | 0.004 | 3.58 | (1.72-7.44) | **0.001*** |

Women who booked ANC late had nearly three times higher odds of experiencing perinatal mortality compared to those who booked early (AOR 2.91, 95% CI: 1.97–4.29, p < 0.001). Maternal anaemia during pregnancy was associated with a three-and-a-half-fold increased risk of perinatal mortality. Women with anaemia had significantly higher odds of perinatal death compared to non-anaemic women (AOR 3.58, 95% CI: 1.72–7.44, p < 0.001). A history of previous pregnancy loss was strongly associated with increased odds of perinatal mortality. Women with a history of stillbirth had five times the odds of perinatal death compared to those without such a history (AOR 5.05, 95% CI: 2.99–8.51, p < 0.001). Table 2 presents a summarized comparison of univariable and multivariable predictors following bivariable and then multivariable logistic regression analysis.

## Discussion

The objective of this study was to identify predictors of perinatal mortality in Lusaka, Zambia. The selected sites are the major hospitals of Lusaka and referral hospitals that anchor all maternal and perinatal health services. Among the socio-demographic variables, educational level and mode of transport were the only variables strongly associated with perinatal mortality. The statistically significant predictors of perinatal mortality identified in this study were mode of transport used by women when accessing healthcare, timing of antenatal care (ANC) bookings, history of abortion, stillbirth, and anaemia during pregnancy.

In clinical settings, pregnancy in women of advanced maternal age (AMA) (i.e., maternal age > 35) is frequently labelled as "high-risk even in the absence of obvious risk factors. This classification has been validated by many authors from a meta-analysis on "advanced maternal age: adverse outcomes of pregnancy", where it was reported that AMA women had higher perinatal mortality and stillbirths compared to women of 20 – 34 maternal age group [36]. On the contrary, maternal age was not found to have significant association with perinatal mortality in this study. Educational level and mode of

transport are the other socio-demographic factors that demonstrated association with our outcome of interest. However, during univariable and multivariable logistic regression analysis only the mode of transport remained statistically significant as a predictor of perinatal mortality. While education level did not show strong association with perinatal mortality in this investigation. A study on effect of maternal education and perinatal outcome done in Punjab, India reported that lower educational status was a significantly important predictor of adverse perinatal outcomes including perinatal mortality [37].

This study revealed that macerated stillbirths and early neonatal deaths (ENND) had a higher occurrence among the perinatal mortality cases, consistent with findings reported in other studies [38,39]. Another study by Shelke et al. [40] similarly reported a higher proportion of macerated stillbirths compared to fresh stillbirths, which may reflect the quality of prenatal and obstetric care, while macerated stillbirths are typically the results of intrauterine foetal death occurring more than 6–12 hours prior to delivery [12,13,41]. Contributing factors to macerated stillbirths include intrauterine growth restriction, placental lesions, and infections, though a large proportion of these deaths remain unexplained [42]. In this study, these cases also remained unexplained, underscoring the need for further investigation into the underlying causes and improved data collection practices. Early neonatal deaths are often caused by complications such as low birth weight, birth asphyxia, and sepsis, which require neonatal intensive care [37,39,43,44]. However, in this study, the causes of early neonatal mortality could not be determined because of the nature of the study design that could measure association.

Among the distal level variables none of them had statistical significance as a predictor of perinatal mortality. On the other hand, it was mainly the intermediate level variables that had significant association with perinatal mortality: mode of transport, ANC booking, and history of stillbirth. Whereas only anaemia during pregnancy was found to be a significant predictor of perinatal mortality among the proximal level variables. The mode of transport showed statistical significance as a predictor of perinatal mortality. This was because women who walked to the healthcare facility to seek medical care during pregnancy were found to be three times more likely to experience perinatal death compared to those with personal transportation. Those using public transport also had two times higher risk of perinatal mortality experience. This finding aligns with a previous study done in Zambia and Uganda, which identified a significant difference in perinatal outcomes based on motorized vs non-motorized means of transport [45]. Factors contributing to this difference may include poor physical access to healthcare, lack of empowerment for decision- making, and limited health education [46]. Furthermore, women who relied on non-motorized transport, such as walking to seek medical care - as was observed in this study - were more likely to access Basic Emergency Obstetric and Neonatal Care (BEmONC) facilities. In contrast, those using motorized transport who had access to Comprehensive Emergency Obstetrics and Neonatal Care (CEmONC) facilities, which provide superior care and improved maternal and perinatal outcomes, particularly when staffed with trained personnel and equipped with essential resources [47,48].

The timing of ANC bookings also played a significant role in perinatal outcomes. The WHO recommends that the first ANC visit should occur within the first trimester, up to 12 weeks of gestation [49]. In this study, over 70% of the women booked late for ANC (> 12 weeks), and late booking was associated with a threefold increased risk of perinatal mortality. This finding corroborated other studies, which reported that delayed ANC increases the risk of complications during pregnancy and delivery, contributing to perinatal mortality [50,51]. A previous history of abortion did not emerge as a significant predictor of perinatal mortality in this study. However, a history of abortion can increase the risk of perinatal mortality compared to those without such a history. This may be attributed to a higher susceptibility to infections during subsequent pregnancies among women with history of abortion [52], which raises the likelihood of perinatal mortality [53]. Conversely, previous history of stillbirth, in this study, was significantly associated with a fivefold increase in the risk of perinatal mortality. This finding aligns with results from other studies in Zambia [54] and Zimbabwe [55], which similarly reported that women with a history of stillbirth faced higher risk of perinatal mortality. This also aligns with findings from a study conducted in Ethiopia [52], which demonstrated that women with a history of stillbirth were at a higher risk of infections during subsequent pregnancies. These infections can compromise foetal health, leading to complications that significantly elevate the likelihood of perinatal mortality [53]. Despite such findings from other studies linking infections to

perinatal mortality, infections were not significantly associated with perinatal mortality in this study. Additionally, many still-births remain unexplained, largely due to the low availability of post-mortem investigations [56]. This lack of clear causality makes discussing stillbirths challenging, not only for the parents but also for healthcare professionals, who often face difficulties in obtaining consent for post-mortem investigation aimed at determining the cause of death [57].

Among the medical complications, anaemia during pregnancy was the strongest predictor of perinatal mortality, with women suffering from anaemia showing a two-and-a-half-fold increased risk, compared to those without anaemia. Anaemia is closely linked to small-for-gestational-age infants and preterm birth, both of which are significant risk factors for perinatal mortality [54]. While previous studies in Zambia [29,58] identified small-for-gestational-age infants, preterm birth, and low birth weight as high-risk factors for perinatal mortality, this study did not find these findings to have significant association with perinatal mortality.

### Study implications

Based on our study findings, socio-demographic factors like lower education level (primary and secondary) and mode of transport showed association with higher likelihood of perinatal mortality, although the latter was not statistically significant in the adjusted model. Participants with personal transport were significantly associated with lower odds of perinatal mortality even in the adjusted multivariable logistic regression model compared to those using public or walked to seek health-care service. This may be suggestive of other factors like lower socioeconomic status and transport accessibility play a significant role in influencing pregnancy outcomes. Further interventions should focus on tackling such socio-demographic issues like improving transport accessibility and education.

The study also highlighted several other factors, such as late ANC booking, a history of stillbirth, and anaemia during pregnancy, which remained significantly associated with higher odds of perinatal mortality even after multivariable regression analysis. These findings imply that health policies should prioritize improving early ANC attendance and targeting high-risk pregnancies, particularly such as anaemia in pregnancy and history of stillbirth should be the centre theme for developing interventions for positive pregnancy outcomes. In this vein, future research should explore potential interventions to reduce the effects of these predictors and improve perinatal outcomes.

### Strengths and limitations

This study provided reasonable assessment of multiple predictors of perinatal mortality simultaneously, offering a nuanced understanding of the factors influencing perinatal outcomes in Lusaka district. The sample is representative of the broader population that has access to the study sites, as data were collected from all major hospitals in the city, ensuring a diverse range of cases that enhances the generalisability of the findings. This broad representation strengthens the relevance of the study to policymakers and healthcare practitioners, as the insights derived can inform targeted interventions and resource allocation in public health strategies. Furthermore, identifying significant predictors not only contributes to the existing body of literature but also equips clinicians with evidence-based knowledge to improve maternal and neonatal care practices.

One notable limitation is the incompleteness of information during data collection, primarily due to inconsistencies in medical record-keeping, and data entry errors which resulted in gaps in some variables. Therefore, some of the variables such as congenital syphilis and congenital anomalies resulted in not being included in the analysis. Additionally, accurately stating or recording the cause of death in stillbirth cases remains challenging, as investigations are not routinely conducted to determine the cause of death. These limitations highlight the need for improved data collection practices and post-mortem investigations to enhance the accuracy and reliability of future research in this area.

### Conclusion

In conclusion, this study identifies key predictors of perinatal mortality in Lusaka, Zambia, including mode of transport, timing of antenatal care initiation, prior history of stillbirths, and maternal anaemia. Women who walked or used public

transport to access healthcare faced higher risks of perinatal mortality, underscoring the socioeconomic disparities affecting access to quality obstetric care. The late initiation of antenatal care compromised early detection and management of complications, emphasizing the need for public health initiatives to promote timely antenatal care. Additionally, history of stillbirth significantly increased the risk of subsequent perinatal mortality, highlighting the necessity for psychological and medical support for women with previous adverse pregnancy outcomes. Maternal anaemia was also a substantial risk factor, correlating with adverse birth outcomes such as small for gestational age and preterm births. While the study did not find small for gestational age or preterm birth to be direct predictors, these findings align with existing literature. Overall, targeted interventions addressing these predictors— such as improving transportation options, enhancing antenatal care accessibility, and providing support for women with prior adverse pregnancy experiences—could significantly reduce perinatal mortality rates and improve maternal and neonatal health outcomes in Lusaka. The aim of this study was to identify the predictors of perinatal mortality in Lusaka, Zambia.

## Recommendations

According to this study findings, to help to reduce the perinatal mortality in Lusaka, Zambia: there is need to improve transportation to enhance options for pregnant women especially in the underserved areas; need to ensure timely access to Comprehensive Emergency Obstetrics and Neonatal Care (CEmONC) facilities; Early ANC booking needs to be promoted through strengthening education and outreach programs to encourage women to book for ANC within the first trimester of pregnancy. In addition, to address anaemia during pregnancy routine screening and treatment need to be strengthened to reduce the associated risks to perinatal mortality. Women with previous experience of perinatal mortality need targeted support and monitoring to minimise risks in subsequent pregnancies. This support includes but not limited to enhanced quality prenatal and obstetrics care investment focusing on early detection and management of pregnancy complications. In addition, mothers with history of pregnancy loss should receive enhanced ANC, including early and routine screening for pregnancy complications, close foetal well-being surveillance, and individualized care plans to address their medical needs. Finally, community-based health education and empowerment programs should be implemented to equip women with knowledge, skills, and resources needed to improve pregnancy outcomes and promote autonomy in decision-making.

## Supporting information

**S1 File. Participant information sheet.**
(S1_File.DOCX)

## Acknowledgments

I would like to thank all my supervisors for the guidance provided to complete this study. I also would like to acknowledge the immense contribution the study participants for accepting to be part of the project.

## Author contributions

**Conceptualization:** Musonda Makasa, Mpundu Makasa.

**Data curation:** Musonda Makasa.

**Formal analysis:** Musonda Makasa.

**Investigation:** Musonda Makasa.

**Methodology:** Musonda Makasa, Patrick Kaonga.

**Project administration:** Musonda Makasa.

**Supervision:** Patrick Kaonga, Choolwe Jacobs, Mpundu Makasa, Bellington Vwalika.

**Writing – original draft:** Musonda Makasa.

**Writing – review & editing:** Musonda Makasa, Mpundu Makasa, Bellington Vwalika.

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
