## [Decision Letter · Decision Letter 0]

19 Jun 2024

PGPH-D-24-01110

Predictors of perinatal mortality in the seven major hospitals of Lusaka Zambia: A Case Control Study

Dear Makasa,

Thank you for submitting your manuscript to PLOS Global Public Health. After careful consideration, we feel that it has merit but does not fully meet PLOS Global Public Health’s publication criteria as it currently stands. Therefore, we invite you to submit a revised version of the manuscript that addresses the points raised during the review process.

We look forward to receiving your revised manuscript.

Kind regards,

Collins Otieno Asweto, PhD

Academic Editor

Journal Requirements:

2. Please provide separate figure files in .tif or .eps format only and remove any figures embedded in your manuscript file. Please also ensure all files are under our size limit of 10MB.

3. In the online submission form, you indicated that "We will make full availability and without restriction all data underlying the findings if requested". 

3. Uploaded as supplementary information.

Reviewers' comments:

Reviewer's Responses to Questions

**Comments to the Author**

1. Does this manuscript meet PLOS Global Public Health’s publication criteria?

Reviewer #1: Yes

Reviewer #2: Partly

2. Has the statistical analysis been performed appropriately and rigorously?

Reviewer #1: Yes

Reviewer #2: No

3. Have the authors made all data underlying the findings in their manuscript fully available (please refer to the Data Availability Statement at the start of the manuscript PDF file)?

Reviewer #1: Yes

Reviewer #2: No

4. Is the manuscript presented in an intelligible fashion and written in standard English?

Reviewer #1: Yes

Reviewer #2: No

Reviewer #1: Dear Authors,

Editorial Office,

This is very valuable contribution on Predictors of perinatal mortality in the seven major hospitals of Zambia.

Methodological framework is sound.

Study was conducted in order to realistically assess the Burden of perinatal mortality in Zambia.

Conclusions are mostly based on results.

Yet I believe the evidence base in insufficiently heterogeneous.

It should be made far more diverse to support claims in the text.

Thus I warmly recommend consideration of inclusion of at least several of beneath suggested published sources alongside with few additional ones at authors own disposal:

https://www.frontiersin.org/journals/public-health/articles/10.3389/fpubh.2022.817717/full?fbclid=IwAR13vrp5Mi5D7gPq11ZDgN5h_d5ay9k2enNhONYPt7SNaol6cbTOxEdn5p8

https://papers.ssrn.com/sol3/papers.cfm?abstract_id=2725386

https://onlinelibrary.wiley.com/doi/full/10.1111/j.1524-4733.2007.00222.x

https://link.springer.com/article/10.1186/s12992-023-00947-4

https://www.mdpi.com/2071-1050/13/19/11038

https://link.springer.com/article/10.1186/s12962-023-00441-z

https://www.frontiersin.org/journals/public-health/articles/10.3389/fpubh.2022.836688/full

https://link.springer.com/article/10.1186/s12961-020-00666-x

https://www.tandfonline.com/doi/full/10.1080/13696998.2021.2007691

https://scidar.kg.ac.rs/handle/123456789/8881

https://www.tandfonline.com/doi/full/10.2147/TCRM.S307587

https://academic.oup.com/alcalc/article/48/4/505/530571

https://link.springer.com/article/10.1186/s12992-018-0348-7

https://www.ajol.info/index.php/ajrh/article/view/260952

https://link.springer.com/article/10.1186/s12916-022-02639-z

https://www.mdpi.com/1660-4601/17/24/9404

http://journals.seedmedicalpublishers.com/index.php/FE/article/view/1220/1488

https://scidar.kg.ac.rs/handle/123456789/8881

https://www.frontiersin.org/journals/public-health/articles/10.3389/fpubh.2017.00023/full

https://www.tandfonline.com/doi/full/10.1080/13696998.2016.1277228

https://www.cell.com/heliyon/fulltext/S2405-8440(24)05581-6

https://www.tandfonline.com/doi/full/10.2147/RMHP.S413630

https://www.mdpi.com/2227-9032/11/10/1507

https://www.tandfonline.com/doi/full/10.2147/RMHP.S388873

https://www.ncbi.nlm.nih.gov/pmc/articles/PMC11127313/

https://link.springer.com/article/10.1186/s12962-024-00521-8

https://dergipark.org.tr/en/pub/bmj/issue/41524/501710

https://www.nature.com/articles/s41599-024-02767-2

https://link.springer.com/article/10.1186/s12962-024-00512-9

https://link.springer.com/article/10.1186/s41256-024-00350-5

https://www.ajol.info/index.php/ajrh/article/view/260952

https://link.springer.com/article/10.1186/s12962-023-00461-9

Kleinman, A. (1982). Neurasthenia and depression: a study of somatization and culture in China. Culture, medicine and psychiatry, 6(2), 117-190.

Conditional to adopting significant share of these recommendation I am willing to reconsider revised manuscript assuming its maturity for publishing.

Reviewer #2: I thank the authors and investigators for documenting this important work. Please find below, comments to improve the manuscript:

Abstract: In the opening statement, please clearly state that the 2.6million babies lost are at global level. State the timeline for this statistic - was this by end of 2015? By end of 2021? For which age is this? Within 28 days of life? Within 7 days of life?

Rephrase the statement, "...... 5.7million since 2000", you probably meant, "5.7 million in 2000".

The statement, "... the rest are in low and middle income countries, 77% in sub-Saharan Africa" is unclear. The rest of what? 77% of what?

State the statistical package and statistical tests used.

There is need for consistency in reporting confidence intervals: one decimal vs two decimal places.

The last sentence of the Conclusion section is unclear: there is need to link the findings on the predictors with the outcome. Also, include a statement on the implication of the study findings.

Regarding the introduction section:

Please clearly define perinatal deaths at the outset. The authors are encouraged to be consistent in the use of the terms "early neonatal death" and "neonatal death" as these have different definitions.

The statement, "Attempts to avoid these perinatal deaths have not yielded much" is ambiguous. The authors are encouraged to use statistics to illustrate the progress Zambia has made in reducing perinatal mortality rate. The SDG annual progress reports contain valuable information that can enrich the argument.

I suggest that the authors rewrite the second half of the first paragraph of the introduction - to maintain the focus on perinatal mortality - first present maternal death as a predictor of early neonatal mortality, before delving into the statistics on maternal mortality. Otherwise, in its current state, the logical flow of ideas on perinatal mortality is distorted.

Revise the last sentence of the introduction section: Additionally, the study also aimed to provide evidence to that guide program interventions to reduce the PMR in this setting." the portion, "... to that guide..." needs re-writing.

In the Methodology section,

What do the authors mean by the statement, "The study did not involve minors"?

Please clarify who the study participants were. Provide a consistent definition of the cases and controls. Were the study subjects, the mothers of the infants or the infants themselves?

Is the Fertility rate of Lusaka similar to the national average of 4.4? You seem to suggest so.

Why was there no attempt to match the controls to the cases, especially by age? Were controls recruited from the same health facility as the cases?

Who provided informed consent? At what point during the postnatal period was data collection done? Was there need to translate the data collection tools?

Under study variables, clearly define the study outcome - perinatal death and the respective attributes. What was the operational definition of a fresh still birth? A macerated still birth? An early neonatal death?

For which of the exposure variables did you obtain data from chart review and from questionnaire interview? How was HIV/syphilis/malaria/tuberculosis/Rhesus/anemia status ascertained? Did you carry out any laboratory tests as part of the study? Findings on some of the variables are not included in the results. For example, what was the relationship between Rhesus status and the study outcome?

Page 10 - first line, what is "congenital louis"? Did you mean "congenital syphilis"? How was this diagnosed? How were congenital abnormalities diagnosed?

Please state the statistical tests used for bivariate analysis.

The statement, "Control was exclusive to the PI for daily monitoring and management in real time." is unclear.

In the results section, it is important that you present the frequencies of the different sub-categories of the outcome: FSB, MSB, ENND? This is included in the second paragraph of the discussion section, which is improper.

Please include a conclusions and recommendations section after the study strengths and limitations.

**Do you want your identity to be public for this peer review?** For information about this choice, including consent withdrawal, please see our Privacy Policy

Reviewer #1: No

Reviewer #2: No

---

## [Decision Letter · Decision Letter 1]

6 Sep 2024

PGPH-D-24-01110R1

Predictors of perinatal mortality in the seven major hospitals of Lusaka Zambia: A Case Control Study

Dear Makasa,

Thank you for submitting your manuscript to PLOS Global Public Health. After careful consideration, we feel that it has merit but does not fully meet PLOS Global Public Health’s publication criteria as it currently stands. Therefore, we invite you to submit a revised version of the manuscript that addresses the points raised during the review process.

We look forward to receiving your revised manuscript.

Kind regards,

Collins Otieno Asweto, PhD

Academic Editor

Journal Requirements:

Reviewers' comments:

Reviewer's Responses to Questions

**Comments to the Author**

Reviewer #3: (No Response)

Reviewer #4: (No Response)

Reviewer #5: All comments have been addressed

publication criteria?

Reviewer #3: Yes

Reviewer #4: (No Response)

Reviewer #5: Yes

3. Has the statistical analysis been performed appropriately and rigorously?

Reviewer #3: Yes

Reviewer #4: (No Response)

Reviewer #5: Yes

4. Have the authors made all data underlying the findings in their manuscript fully available (please refer to the Data Availability Statement at the start of the manuscript PDF file)?

Reviewer #3: No

Reviewer #4: (No Response)

Reviewer #5: Yes

5. Is the manuscript presented in an intelligible fashion and written in standard English?

Reviewer #3: Yes

Reviewer #4: No

Reviewer #5: Yes

Reviewer #3: This is an excellent manuscript that has focused on an important topic for the wellbeing of pregnant women. The findings will help to inform policy makers in Zambia on how to improve the quality of antenatal care and the need for targeted structural interventions that will address transport issues and other social determinants of health. There are few areas which I would like to suggest to the authors to work on in order to further improve the manuscript before it is published. The areas for improvement are as follows:

ABSTRACT

Results

The sentence: “had almost three times odds of experiencing perinatal (AOR 2.909, 95% CI: 1.97-4.29), p <0.001”; I suggest to the authors to INSERT mortality between perinatal and the bracket.

The sentence: “perinatal mortality than to those who had not (AOR 5.047, 95% CI: 2.99-8.51). Conclusion: This”; I suggest to the authors to DELETE "to" between than & those.

The sentence: “facility, anaemia in pregnancy, and previous history of loss of baby before birth perinatal death”; I suggest to the authors to DELETE "perinatal death" after birth.

INTRODUCTION

First paragraph:

o Line eight (until recently most stillbirths not accounted for in the worldwide data tracking, while no social), I suggest to the authors to INSERT "were" between stillbirth & not.

o Line ten (The Maternal Mortality Estimated Interagency Group (MMEIG) reported an estimated 295,000), I suggest to the authors EDIT “Estimated” in the sentence so that the sentence will be as follows “The Maternal Mortality Estimation Inter-Agency Group (MMEIG)”.

o Line eleven (maternal deaths globally in 2017, and 196,000 (66%) were from sub-Saharan Africa (SSA) (4).), I suggest to the authors to REPLACE "and" with "in which" between 2017, & 196,000(66%).

Second paragraph:

o Lines 10-13 (mortality. SSA has the highest Perinatal Mortality Rate (PMR) (42.95 per 1000 live births) with Nigeria leading followed by Ethiopian 40.9 and 49 per 1000 live births respectively, whereas southern Africa is approximately 30.3 according to systematic reviews and meta-analyses by Akombi and Renzaho (10), (11).). I have the following observation and a suggestion:

The Data for Ethiopia is not correct. The paper by Akombi and Renzaho (Akombi BJ and Renzaho AM. Perinatal Mortality in Sub-Saharan Africa: A Meta-Analysis of Demographic and Health Surveys. Annals of Global Health. 2019; 85(1): 106, 1–8. DOI: https://doi.org/10.5334/aogh.2348 ) https://www.ncbi.nlm.nih.gov/pmc/articles/PMC6634369/pdf/agh-85-1-2348.pdf indicates that "Lesotho is the one with the highest rate of 49.6 then Nigeria follows with 40.9

Therefore, I suggest to the authors to look on that and make the necessary corrections.

Third paragraph:

o Second line (figures. The PMR for Zambia is 33 per 1000 population (12), which is above the SSA average and);

I suggest to the authors to look on that sentence again and the paper source because the information “which is above the SSA average” is not correct.

The SSA rate is 34.7 per 1000 live births as per the paper by Akombi and Renzaho (Akombi BJ and Renzaho AM. Perinatal Mortality in Sub-Saharan Africa: A Meta-Analysis of Demographic and Health Surveys. Annals of Global Health. 2019; 85(1): 106, 1–8. DOI: https://doi.org/10.5334/aogh.2348 ) https://www.ncbi.nlm.nih.gov/pmc/articles/PMC6634369/pdf/agh-85-1-2348.pdf

MATERIALS AND METHODS

Last sentence (These facilities also serve the most densely populated populations in the country.);

o Based on the context of the paragraph, I suggest to the authors to replace “populations” with “districts”.

STUDY DESIGN

Fifth line (days of life from from the study sites. Whereas, two controls for every case underwent systematic),

o I suggest to the authors to DELETE the repeating word "from".

Line eight (neonatal deaths controls were those who delivered at least within 24hrs of the date of birth of of),

o I suggest to the authors to DELETE the repeating word "of".

Nineth line (the this case. A midwife from each labour ward and postnatal ward from each facility were enrolled),

o I suggest to the authors to DELETE "this" between “the” & “case”.

STUDY POPULATION

Figure 1.0 Selection of study participants’ flow chart – second box (Eligible population);

o I suggest to the authors to EDIT “with 24 hours” to be "within 24 hours".

SAMPLE SIZE CALCULATED

I suggest to the Authors to edit the above heading to be “Sample Size Calculation”.

RESULTS

Participants’ socio-demographic characteristics

Lines 5-6 (between 1 to 4 while the grandmultipara were 98 (17.3%). 445 (72.4%) represented the unemployed and 170 (27.6%) unemployed. In terms of religious background, respondents);

o I suggest to the authors to relook at the two lines because it is not clear which percentage is for employed and unemployed.

Predictors of perinatal mortality from bivariate analysis

Fourth line (when going to the hospital also had association (on foot) with perinatal mortality (p<0.001).);

o I suggest to the authors to MOVE (on foot) to be after hospital (between hospital & & also).

Unadjusted logistic regression of variables associated with perinatal mortality

First paragraph

o Fifth line (slightly over 3 and almost twice times (OR=3.482, 95% CI: 1.87-6.49) and (OR=1.964, 95% CI:);

I suggest to REPLACE twice with 2.

o Seventh line (transport respectively, comparison to those with personal transport to access healthcare services.);

I suggest to the authors to EDIT comparison to be compared.

o Twelveth line (History of abortion had three time likely (OR=3.102, 95% CI: 1.16-8.28) to experience perinatal);

I suggest to the authors to REPLACE had with were.

Second paragraph

o Second line (control for confounding, it required multivariable logistic regression analysis. The above table is);

I suggest to the authors to REPLACE "The above table" with "Table 1.0".

o Nineth line (half times more associated with perinatal mortality than those without anaemia did in pregnancy):

I suggest to the authors to DELETE "did" between anaemia & in.

DISCUSSION

Second paragraph

o Seventh line (Maceration onset can range from 6 – 12 hours. Factors that contribute to macerated stillbirths);

I suggest to the authors to INSERT "to" between 6 and 12.

Third paragraph

o Second line (medical care during pregnancy has three times more likely to suffer perinatal demise than those);

I suggest to the authors to REPLACE "has" with "were" between pregnancy & three.

o Twelveth line (commences (28). Another plausible reason is according the study in Uganda and Zambia by Sacks,);

I suggest to the authors to INSERT "to" between according & the.

Fifth paragraph:

o First line (Women who had had a history abortion before also had three time more at risk of having a perinatal);

I suggest to the authors to INSERT "of" between history & abortion; and REPLACE had with “were” between also & three.

o Thirteenth line (they endeavour to obtain consent for investigations to try ascertain the cause of death (39));

I suggest to the authors to INSERT "to" between try & ascertain".

RECOMMENDATIONS

First five lines (According to this study findings, in order to help to reduce the perinatal mortality in Lusaka, Zambia. There is need to improve transportation to enhance options for pregnant women, especially in the underserved areas, to ensure timely access to CEmONC facilities. Early ANC booking needs promoted to strengthen education and outreach programs to encourage women to book for ANC within the first trimester of pregnancy. In);

o I suggest to the authors to REPHRASE THIS TO BE: According to this study findings, in order to help to reduce the perinatal mortality in Lusaka, Zambia: there is need to improve transportation to enhance options for pregnant women especially in the underserved areas; need to ensure timely access to CEmONC facilities; Early ANC booking needs to be promoted through strengthening education and outreach programs to encourage women to book for ANC within the first trimester of pregnancy.

Sixth and seventh lines (during pregnancy routine screening and treatment for anaemia during pregnancy needs strengthened approaches to reduce associated risks to perinatal mortality. Women with previous):

o I suggest to the authors to INSERT “to be” between needs & strengthened.

Twelveth and thirteenth lines (programs to equip women with knowledge and resources for better pregnancy outcomes and self autonomy in decision-making.);

o I suggest to the authors to INSERT "should be strengthened" between programs & to.

Reviewer #4: REVIEWER REPORT

TOPIC

Predictors of perinatal mortality in the seven major hospitals of Lusaka Zambia: A Case Control Study

Remark(s) 1: Reads fine.

ABSTRACT

Methods: this was

Remark(s) 2: The article “this” should read “This”.

Keywords: Perinatal, proportions, predictors, mortality

Remark(s) 3: Replace the word “proportion” with another word which more directly related to the study.

INTRODUCTION

Page 3, para 1, sentence 3: “Attempts to avoid these perinatal deaths have not yielded much.”

Remark(s) 4: Replace the word “avoid” with either “prevent” or “eliminate”.

Page 3, para 1, sentence 4: “For example, until recently most stillbirths not accounted for in the worldwide data tracking, while social recognition and lack of investment and programmatic action have contributed to this problem (2).”

Remark(s) 5: The sentence does not read well; the meaning is lost. Recast.

Page 3, para 1: “Literature shows a strong linkage between maternal deaths and perinatal mortality….”

Remark(s) 6: Which literature? Provide source(s) for this and it should be right after the first word (literature).

Page 3, para 1: “Almost two thirds of maternal death causes also account for the causes of perinatal deaths (6).”

Remark(s) 7: This does not read well. You may consider: “Almost two thirds of the causes of maternal death also account for perinatal deaths (6).”

Page 3, para 1, last sentence: “Whereas early neonatal mortality is a subset of perinatal mortality and refers to loss of newly born within 7 days of life (7) and for purposes epidemiological studies and statistics (8).”

Remark(s) 8: This does not read well, recast.

Page 3, para 2, sentence 1: “The World Health Organization Sustainable Development Goal (SDG) number 3.2 targets to end….”

Remark(s) 9: This should rather read: “The World Health Organization’s Sustainable Development Goal (SDG) 3.2 targets to end….”

Page 3, para 2, last sentence: “The neonatal mortality rate also demonstrated a downward trend from 37 in 1990 to 19….”

Remark(s) 10: Replace the word “demonstrated” with either “showed” or “revealed”.

Page 4, para 1, sentence 2: “These strides however, have only been realistic in High-Income Countries….”

Remark(s) 11: Replace the word “realistic” with “possible”.

Page 4, para 1, last sentence: “Africa is approximately 30.3 according to systematic reviews and meta-analyses by Akombi and Renzaho (12), (13).”

Remark(s) 12: Are references (12, 13) authored by Akombi and Renzaho? If so, then the reference must read: “Akombi and Renzaho (12, 13)”. If not, then cite reference (13) too as a non-parenthetical reference.

Page 4, para 2, sentence 2: “The PMR for Zambia is 33 per 1000 population (14), which is above the SSA average and therefore still far from attaining the SDG and Vision 2030 set targets (15).”

Remark(s) 13: Replace the word “attaining” with “meeting”.

MATERIALS AND METHODS

Page 4, para 1, sentence 3: “Whereas, the participants were prospectively recruited as and when cases occurred and if agreeable to partake of the research process.”

Remark(s) 14: This is not clear, recast.

Page 4, para 1, sentence 4: “Information about the research was share with the ….”

Remark(s) 15: Incorrect tense; the word “share” should read “shared”

Page 5, para 1, last sentence: “…from all study facilities during period January to December 2023.”

Remark(s) 16: Omission; insert the article “the” between the words “during” and “period”.

Study Design

Page 5: “After sharing the participant information sheet and obtaining an informed consent.”

Remark(s) 17: This is incomplete; recast.

Study population

Page 5, sentence 2: “Refer to the following figure 1.0 flowchart that illustrates the source of the sample, eligibility screening process and how the final sample was arrived at.”

Remark(s) 18: This does not read well; rather consider this: “Figure 1.0 illustrates the source of the sample, eligibility screening process and how the final sample was arrived at.”

Inclusion criteria

Page 7, sentence 1: “Cases included stillbirths with mothers above 18 years of age that were delivered with a gestational age above 24 weeks of gestation.”

Remark(s) 19: Repetition; delete the phrase “of gestation” from the sentence.

Exclusion criteria

Page 7: “Women who chose to withdraw from the study, below the age of 18 as they could not sign for consent. Any pregnancy below 24 weeks of gestation or in the absence of gestational age birthweight less than 500g. Women who chose not to participate after sharing information from the participant information sheet.”

Remark(s) 20: This is incorrect, as none of the above categories do not meet the study objectives of the study and do not qualify to participate in the study. These ones are already excluded from the study and therefore cannot be part of your exclusion criteria. Typically, the exclusion criteria must cover participants who qualify in every way to be part of the study but for one reason or the other cannot participate or be included in the study. These may be due to the absence of the participants during the study period, inability of the participant to be part of the study due to ill health or other factors such as being too emotional to provide accurate information. The authors must redefine the exclusion criteria.

Sampling technique

Page 9: “Sample size per facility based on probability proportional to size sampling after reviewing the previous year records from January to December 2023. However, for analysis only 630 underwent analysis while the remainder did not meet the inclusion criteria. Upon enrolment of a case, two controls randomly selected from within the 24 hours shift that a case occurred. The disparity in the number since it was a 1:2 selection is because of exclusion during data cleaning and some entry errors.”

Remark(s) 21: This sub-section does not read well as most of the sentences appear hanging and incomplete. The sub-section must be rewritten completely. The authors should consult an English Language expert for proofreading before resubmission.

Data collection tool

Page 9: “A standard interviewer administered questionnaire was set….”

Remark(s) 22: The nomenclature: “interviewer administered” may appear misleading to the global audience as it could be misconstrued for a qualitative approach, whereas the study is actually using a quantitative approach. Rather, I suggest you go with: “researcher administered”.

Data analysis

Page 10: “Descriptive statistical analyses summarized of proportions and frequency of cases and controls. Univariable analysis determined the crude association between perinatal mortality and independent variables. To demonstrate association with a p-value of <0.05, multivariable logistic regression analysis to show the association. Some continuous variables such as age, weight, parity, and gravida introduction into the model as categorical variable was on intuitive from a clinical standpoint.”

Remark(s) 23: These sentences do not read well; recast. Make sure your sentences are complete and not hanging as in the above.

Ethical considerations

Page 11: “There was Confidentiality and privacy for study participants.”

Remark(s) 24: Provide further details on how specifically this was ensured.

RESULTS

Predictors of perinatal mortality from bivariate analysis

Page 12: “Bivariate analysis exposed several variable associated with perinatal mortality.”

Remark(s) 25: This should rather read: “Bivariate analysis revealed several variables associated with perinatal mortality.” There seems to be too many omissions and grammatical issues throughout the paper. I strongly suggest that the authors consult an English Language expert to proofread the work before it is resubmitted.

Unadjusted logistic regression of variables associated with perinatal mortality

Page 12: “According to the univariate and multivariable logistic regression for factors associated with perinatal death in table 5.”

Remark(s) 26: This sentence is incomplete. There are just too many of such throughout the paper.

DISCUSSION

Page 16, para 2: “This is in line with some previous studies that reported results of a similar nature (22).”

Remark(s) 27: You indicated that the finding is consistent with some previous studies yet you provided only one citation; fix this.

Page 16, para 2: “The authors attributed this to be a reflection of the quality of prenatal and obstetric care, with higher fresh to macerated ratios implying poorer care.

Remark(s) 28: This does not read well; recast.

Remark(s) 29: The paragraph is over focused on what other studies found rather than focusing on what the findings of the current study. This is not good discussion as the global audiences would rather be interested in reading more about your findings and how they compare with other studies. Rewrite the paragraph by focusing more on the findings of the current study.

Page 17, para 2: “Women who walked to the facility to seek medical care during pregnancy has three times….”

Remark(s) 30: There is subject verb disagreement; fix. There are other examples in the paper; fix this throughout the paper.

Page 17, para 2: “This finding is in tandem with a previous study done in Zambia and Uganda that reported motorized vs non-motorized means of transport to have significant difference statistically (29).”

Remark(s) 31: Recast this to read: “This finding is in tandem with a previous study done in Zambia and Uganda (29) that reported….” Let this guide you in similar instances throughout the paper.

Remark(s) 32: As indicated in “Remark(s) 29” above, the paragraph 3 is also too focused on what other studies found rather than focusing on what the findings of the current study. Additionally, where the discussion is about the current study, the authors must clearly indicate that by stating “findings of the current study”. This would help distinguish findings of the current study from previous ones. Rewrite the paragraph by focusing more on the findings of the current study.

Page 18, para 1: “…late ANC initiation had three times the risk of perinatal mortality compared to those who started early (<12 weeks gestation)….”

Remark(s) 33: This is not well written. Recast to read: “…women with late ANC initiation have three times the risk of perinatal mortality compared to those who started early (<12 weeks gestation)….”

Page 18 & 19, para 1: “This experience is supported by other studies too that reported on early antenatal care failure resulting in potential complications during pregnancy, delivery, and puerperium that inadvertently increase risk of perinatal mortality (34, 35).

Remark(s) 34: This should read: “This experience is supported by other studies (34, 35) that reported….”

Page 19, para 1: “Women who had had a history abortion before also had three time more at risk of having a perinatal death compare to the women who had not.”

Remark(s) 35: There are omissions here; this should read: “Women who had had a history of abortion before also had three times more at risk of having a perinatal death compare to the women who had not.

Page 19, para 1: “A recent study on stillbirths determinants reported similar findings of higher risk of stillbirth among those with prior experience of stillbirth (34). This is also in line with another similar study by Dube, Lavender (35) in Zimbabwe that reported similar findings.”

Remark(s) 36: The first sentence should read: “A recent study (34) on the determinants of stillbirths reported similar findings of higher risk of stillbirth among those with prior experience of stillbirth.

The second sentence should read: “Furthermore, findings of the current study also affirm a previous study by Dube and Lavender (35) in Zimbabwe that reported similar findings.”

Page 19, para 1: “History of abortion in this study demonstrated statistical significance as predictor of perinatal mortality.”

Remark(s) 37: There are issues with this. First, what do you mean by “in this study”? Do mean the current study or Dube and Lavender (35) in the previous sentence. There are real issues with the organisation and presentation of the discussion.

Page 19, para 1: “The finding is consistent with a study done in Ethiopia on the effects of previous stillbirth or abortion on subsequent pregnancies and infants increased the risk of infections (36), which increase the risk of perinatal mortality (37).”

Remark(s) 38: I struggling to understand this. The authors must seek help from an experienced researcher. There are too many issues with paper.

Remark(s) 39: I strongly recommend that the whole section (discussion) be rewritten given the volume of errors found. The authors would seriously need help from both an English Language expert and an experienced researcher to proofread the manuscript before it is resubmitted.

STRENGTHS AND LIMITATIONS

Page 20: “The study was able to investigate…. In this investigation’s setting, the sample.... The identification of significant predictors provides evidence to not only public health and policy makers but to clinical practice too. Stating or recording cause of death is difficult in stillbirths because investigations are not routine in an endeavour to assign cause of death at least.”

Remark(s) 40: There are real issues with the section. First, the study CANNOT investigate (The study was able to investigate). Second, most of the sentences do not read well and thus hard to understand.

CONCLUSION

Page 20: “The aim ….

Remark(s) 40: This is woefully inadequate.

RECOMMENDATIONS

Pages 20 & 21: “According to this study findings, in order to help to reduce the perinatal mortality in Lusaka, Zambia. There is need to.…”

Remark(s) 41: There are issues with this section. First, the first sentence does not communicate anything. Second, the recommendations were NOT directed at any specific individual or organization for action. This is not good enough. The authors need to fix this.

COMMENTS FOR THE AUTHORS

I commend you for the good effort at addressing an area of public that is so dear to me and most public health experts. However, there are real issues with the paper that will require much effort to fix. I will suggest you seek help from both an English Language expert and experienced researcher to conduct a thorough proofreading before you consider a resubmission.

Reviewer #5: Great work has been done by the authors on the manuscript and a lot of improvement has been made.

Your objectives are clearly presented besides a comprehensive background to the study. data analysis which is very important appears to have been done thoroughly. However, the manuscript still needs some important improvement. The major one is that there are many grammatical issues in the manuscript that requires editing. I will suggest the authors get a professional editing done on the manuscript to improve its overall readability. In the exclusion criteria clarify the real target of the study is; mothers, stillbirths or neonates? Or mothers were just proxies to get to the stillbirths, neonates? Clarification is also needed in terms of your data collection too. You indicated that you used standard questionnaire. Was this tool adapted or adopted and was this tool pretested prior to data collection? Please, clarify.

Overall, this is very important research, and great effort can be seen exerted in it.

**Do you want your identity to be public for this peer review?** For information about this choice, including consent withdrawal, please see our Privacy Policy

Reviewer #3: **Yes: ** Eliudi Saria Eliakimu

Reviewer #4: **Yes: ** BOTHA, Nkosi Nkosi

Reviewer #5: No

---

## [Decision Letter · Decision Letter 2]

4 Nov 2024

PGPH-D-24-01110R2

Predictors of perinatal mortality in the seven major hospitals of Lusaka Zambia: A Case Control Study

Dear Makasa,

Thank you for submitting your manuscript to PLOS Global Public Health. After careful consideration, we feel that it has merit but does not fully meet PLOS Global Public Health’s publication criteria as it currently stands. Therefore, we invite you to submit a revised version of the manuscript that addresses the points raised during the review process.

We look forward to receiving your revised manuscript.

Kind regards,

Collins Otieno Asweto, PhD

Academic Editor

Journal Requirements:

Reviewers' comments:

Reviewer's Responses to Questions

**Comments to the Author**

Reviewer #4: All comments have been addressed

Reviewer #6: (No Response)

publication criteria?

Reviewer #4: Yes

Reviewer #6: Yes

3. Has the statistical analysis been performed appropriately and rigorously?

Reviewer #4: Yes

Reviewer #6: No

4. Have the authors made all data underlying the findings in their manuscript fully available (please refer to the Data Availability Statement at the start of the manuscript PDF file)?

Reviewer #4: Yes

Reviewer #6: Yes

5. Is the manuscript presented in an intelligible fashion and written in standard English?

Reviewer #4: Yes

Reviewer #6: Yes

Reviewer #4: There are still few corrections that must be done. For instance, under the introduction on page 4, para 1, last sentence, references 13 and 14 must be in one parenthesis. Also, on page 11 under ethical consideration, sentence 4, the 8th word "purposed" should read "purpose". There are other omissions and commissions that must be fixed.

Reviewer #6: Review comment: Predictors of perinatal mortality in the seven major hospitals of Lusaka Zambia: A Case Control Study

Methodology:

Study design:

o Precise if the study design was prospective case control.

o Take this sentence in sampling technique:” due to rarity of the outcome purposeful sampling was used for the case that met the inclusive criteria.”

Sampling Technique:

o Put more information on how you used PPS: Probability Proportional to size sampling: was this technique used to select major hospital in the city or participants: if it was used to select major hospitals just show how it was applied. Then for the issue of representativity why not using proportionate stratified random sampling for selecting people from 7 hospitals:

o The final selected participants must correlate with the final sample size and the findings total.

o Your final selected participants were 630 Study participants, your final sample size is 558 of study participants and the total of findings are also showing different results. Just try to match these different parts for example apply the sample correction factor or explain why you have taken more study participants and if there were missing participants in findings.

Data analysis:

o Put the chi-square test used in data analysis, write also that all factors associated with perinatal death in bivariate analysis (Chi-square) were transferred to Logistic regression for confounder adjustment.

o Avoid the confusion of the word univariate analysis and bivariate analysis. Because univariate is for descriptive analysis.

Findings:

o The logistic regression can use only crude odd ratio: Bivariate and Adjusted Odd ratio (for multivariate). Just remove the confusion in your commentary.

**Do you want your identity to be public for this peer review?** For information about this choice, including consent withdrawal, please see our Privacy Policy

Reviewer #4: **Yes: ** BOTHA Nkosi Nkosi

Reviewer #6: **Yes: ** Dr. Nsanzabera Charles MPH, PhD.

---

## [Decision Letter · Decision Letter 3]

7 Jan 2025

PGPH-D-24-01110R3

Predictors of perinatal mortality in the seven major hospitals of Lusaka Zambia: A Case Control Study

Dear Makasa,

Thank you for submitting your manuscript to PLOS Global Public Health. After careful consideration, we feel that it has merit but does not fully meet PLOS Global Public Health’s publication criteria as it currently stands. Therefore, we invite you to submit a revised version of the manuscript that addresses the points raised during the review process.

We look forward to receiving your revised manuscript.

Kind regards,

Collins Otieno Asweto, PhD

Academic Editor

Journal Requirements:

Reviewers' comments:

Reviewer's Responses to Questions

**Comments to the Author**

Reviewer #6: (No Response)

Reviewer #7: (No Response)

publication criteria?

Reviewer #6: Yes

Reviewer #7: Yes

3. Has the statistical analysis been performed appropriately and rigorously?

Reviewer #6: Yes

Reviewer #7: Yes

4. Have the authors made all data underlying the findings in their manuscript fully available (please refer to the Data Availability Statement at the start of the manuscript PDF file)?

Reviewer #6: Yes

Reviewer #7: Yes

5. Is the manuscript presented in an intelligible fashion and written in standard English?

Reviewer #6: Yes

Reviewer #7: Yes

Reviewer #6: Reviewer comment: Predictors of perinatal mortality in the seven major hospitals of Lusaka Zambia: A Case Control Study

Methodology:

All factors found to be associated with perinatal mortality in the bivariate analysis (p-value <0.05): add at 95% confidence interval.

Findings:

Review the title of Table 1.0: Summary of the maternal socio-demographic characteristics frequency distribution. it is not only socio-demographic characteristics. There are actually other factors such as comorbidities, lifestyle such as smoking and alcohol use. Review the title or split it into different tables.

In table 1.0: missing percentage:

ANC Booking: one line misses percentage 106 (%)

Preeclampsia: misses percentage 160(%)

It would have been better to add total on each variable to track the missing values on each variable and checking if the total results correlated with included final sample size.

Discussion:

Discuss this study implication??

Reviewer #7: This paper investigates predictors of perinatal mortality in Lusaka, Zambia, using a multi-center case control approach of 210 cases of perinatal death compared with 420 controls. The authors found that late antenatal care visits after 12 weeks gestation, walking to the hospital instead of using personal transport, maternal anemia, and a history of pregnancy loss significantly increased the risk of perinatal mortality. The results identify critical areas for intervention, including improving access to timely antenatal care and addressing maternal health issues to reduce perinatal mortality in low-income countries.

The study is well-designed with compelling results on an important topic, making it a strong candidate for publication. It has the potential to meaningfully impact practices in low- and middle-income countries.

Recommendations:

Intro:

1. Consider streamlining the statistics presented in the introduction to improve readability. For example, while the statistics on maternal deaths and the absolute numbers of global perinatal mortality are noteworthy, they may be less directly relevant to the focus of this paper. Instead, incorporating a statistic on the current global neonatal mortality rate and its trends over time could provide context for the reader especially for comparison with the WHO sustainable development goals.

2. Consider providing a definition and brief overview of maceration in the introduction, and cite relevant references for the readers’ understanding.

Methods:

3. Consider specifying the total number of hospitals in Lusaka and describe the criteria used to select the facilities included in the study. Include an estimation of the number of births per year in Lusaka.

4. Please provide details on how study variable data was collected: through chart review? Through maternal interview?) If different variables were collected using different methods, please specify which were collected using what method.

5. Please provide details on the five participants who were excluded.

6. Please clarify whether any infants greater than 24 weeks of gestation weighed less than 500 grams.

7. Include information about Group B Strep status and the duration of rupture of membranes (ROM) if available. If not, consider discussing the absence of this data in the discussion section.

8. Please provide details on data collection for the 7-day follow-up of the control group. Specifically, clarify how it was determined that the controls were healthy during this period.

9. Please include the researcher questionnaire as a supplemental file for transparency and reproducibility.

Results:

10. For Table 1, the percentages should be presented as intragroup values (e.g., percentages within cases for age groups ≤19, 20–34, >35, and similarly for controls) rather than intergroup comparisons (cases vs. controls) for clarity. Additionally, please include the sample size (N) at the top for both cases and controls. “Folate and Ferrous sulphate” heading should clarify that it is supplementation: “Folate and Ferrous sulphate supplementation”

11. Please clarify whether "malaria" refers to a history of malaria (and if so, whether this was during pregnancy) or active malaria at the time of delivery. If data is available on the timing of malaria infection (ie, which trimester of pregnancy it occurred), please include this information.

12. Please include proximal level variables and their comparison in Table 1, consider including text in the results section as well.

13. The description of the bivariate analysis using the chi-square test should be moved to the Methods section for consistency.

14. Analyzing macerated vs. non-macerated cases could be interesting and provide additional insights into variables affecting stillbirth. Consider conducting a comparative subgroup analysis to identify the variables associated with each category.

15. Consider further stratifying the analysis by hospital type or care level (tertiary vs first level hospitals) to explore potential variations across different healthcare settings.

16. Please include data on the average number of days for early neonatal mortality in cases to provide additional context.

Other comments:

A thorough copy-editing of the manuscript is recommended to address minor spelling and grammatical errors. Some specific (but not comprehensive) examples include:

• In the Abstract Methods section of the abstract, capitalize the first word (“This”).

• In the Abstract Results, revise the sentence to read: “...12 weeks gestation had almost three times the odds of experiencing perinatal mortality.” Additionally, place the p-value within the parentheses.

• In the same section, revise the sentence to read: “Walking as a means of reaching the healthcare facility had over three times the odds of perinatal mortality.”

• In the Introduction, the abbreviation “PMR” is introduced twice; only the first instance needs to be kept.

• In the Study Design section, correct the repetition in the third line: “from from.”

• In the Methods, revise “controls were ‘follow up’ before discharge” to “controls were followed up before discharge.”

• In the Inclusion Criteria section, correct the formatting to avoid having two periods after the first sentence.

• In the Patients’ Socio-Demographic Characteristics section, capitalize the first letter of religions.

• In the same section, when discussing socio-demographic characteristics, avoid simply listing the numbers; either include them in parenthesis or include a description.

Overall, this paper provides valuable insights into the factors influencing perinatal mortality in low- and middle-income countries and has the potential to significantly contribute to the literature.

**Do you want your identity to be public for this peer review?** For information about this choice, including consent withdrawal, please see our Privacy Policy

Reviewer #6: **Yes: ** Nsanzabera Charles

Reviewer #7: No

---

## [Decision Letter · Decision Letter 4]

14 Feb 2025

PGPH-D-24-01110R4

Predictors of perinatal mortality in the seven major hospitals of Lusaka Zambia: A Case Control Study

Dear Makasa,

Thank you for submitting your manuscript to PLOS Global Public Health. After careful consideration, we feel that it has merit but does not fully meet PLOS Global Public Health’s publication criteria as it currently stands. Therefore, we invite you to submit a revised version of the manuscript that addresses the points raised during the review process.

We look forward to receiving your revised manuscript.

Kind regards,

Collins Otieno Asweto, PhD

Academic Editor

Journal Requirements:

Reviewers' comments:

Reviewer's Responses to Questions

**Comments to the Author**

Reviewer #4: All comments have been addressed

Reviewer #6: All comments have been addressed

Reviewer #8: All comments have been addressed

publication criteria?

Reviewer #4: Yes

Reviewer #6: Yes

Reviewer #8: Yes

3. Has the statistical analysis been performed appropriately and rigorously?

Reviewer #4: Yes

Reviewer #6: Yes

Reviewer #8: Yes

4. Have the authors made all data underlying the findings in their manuscript fully available (please refer to the Data Availability Statement at the start of the manuscript PDF file)?

Reviewer #4: Yes

Reviewer #6: No

Reviewer #8: No

5. Is the manuscript presented in an intelligible fashion and written in standard English?

Reviewer #4: Yes

Reviewer #6: Yes

Reviewer #8: Yes

Reviewer #4: All corrections effected to my satisfaction.

Reviewer #6: Review report:

Abstract:

Standardize the decimals.

Findings:

i. Table1.0: Please include totals this will clarify your results and help a good digestion and understanding of your results.

ii. Check your totals are not reflecting your (n)

iii. Standardize your decimals in the table and in all your results in abstract and tables.

Discussion:

i. Discuss your study socio-demographic objectives.

ii. Try to compare and contrast your findings in understandable manner: for example, this paragraph:(Furthermore, women using non-motorized transport were likely to access only Basic Emergency Obstetric and Neonatal Care (BEmONC) facilities, compared to those using motorized transport who had access to Comprehensive Emergency Obstetrics and Neonatal Care (CEmONC) facilities (40). As the latter offer superior care for pregnancy complications especially if they have trained staff and are better equipped (41)) is not well explained how it is related to your findings. At which point you referred? is it high or low compared to your study?

iii. Discuss the study implications

iv. Ensure punctuation is well done in your discussion.

Reviewer #8: All comments have been addressed.

**Do you want your identity to be public for this peer review?** For information about this choice, including consent withdrawal, please see our Privacy Policy

Reviewer #4: **Yes: ** BOTHA, Nkosi Nkosi

Reviewer #6: No

Reviewer #8: No

---

## [Decision Letter · Decision Letter 5]

9 Apr 2025

PGPH-D-24-01110R5

Predictors of perinatal mortality in the seven major hospitals of Lusaka Zambia: A Case Control Study

Dear Dr. Makasa,

Thank you for submitting your manuscript to PLOS Global Public Health. After careful consideration, we feel that it has merit but does not fully meet PLOS Global Public Health’s publication criteria as it currently stands. Therefore, we invite you to submit a revised version of the manuscript that addresses the points raised during the review process.

We look forward to receiving your revised manuscript.

Kind regards,

Collins Otieno Asweto, PhD

Academic Editor

Journal Requirements:

Additional Editor Comments (if provided):

Reviewer's Responses to Questions

**Comments to the Author**

Reviewer #6: All comments have been addressed

Reviewer #7: (No Response)

Reviewer #9: (No Response)

Reviewer #10: (No Response)

Reviewer #11: (No Response)

Reviewer #12: All comments have been addressed

publication criteria?

Reviewer #6: Yes

Reviewer #7: Yes

Reviewer #9: Yes

Reviewer #10: Yes

Reviewer #11: Yes

Reviewer #12: Yes

3. Has the statistical analysis been performed appropriately and rigorously?

Reviewer #6: Yes

Reviewer #7: Yes

Reviewer #9: Yes

Reviewer #10: Yes

Reviewer #11: Yes

Reviewer #12: Yes

4. Have the authors made all data underlying the findings in their manuscript fully available (please refer to the Data Availability Statement at the start of the manuscript PDF file)?

Reviewer #6: Yes

Reviewer #7: Yes

Reviewer #9: Yes

Reviewer #10: Yes

Reviewer #11: (No Response)

Reviewer #12: Yes

5. Is the manuscript presented in an intelligible fashion and written in standard English?

Reviewer #6: Yes

Reviewer #7: Yes

Reviewer #9: Yes

Reviewer #10: Yes

Reviewer #11: (No Response)

Reviewer #12: Yes

Reviewer #6: All comments were addressed

Reviewer #7: This paper investigates predictors of perinatal mortality in Lusaka, Zambia, using a multi-center case control approach of 210 cases of perinatal death compared with 420 controls. The authors found that late antenatal care visits after 12 weeks gestation, walking to the hospital instead of using personal transport, maternal anemia, and a history of pregnancy loss significantly increased the risk of perinatal mortality. The results identify critical areas for intervention, including improving access to timely antenatal care and addressing maternal health issues to reduce perinatal mortality in low-income countries.

The study is well-designed with compelling results on an important topic, making it a strong candidate for publication. It has the potential to meaningfully impact practices in low- and middle-income countries. Overall, the authors have addressed reviewer comments effectively, with only minor suggestions remaining.

Recommendations:

Methods:

1. Is there an estimation of how many births per year in Lusaka, or how many births per year at each of the hospitals? If so, please include this information.

2. Consider including the reason 5 participants were excluded in the flow chart or methods: “withdrawn by mother due to emotional distress”

3. Please clarify details on 7-day follow up of control group. Were infants seen in a postnatal clinic at 7 days?

Below are stylistic suggestions to improve clarity of the Introduction and Methods for the reader.

The introduction provides valuable background information, but could benefit from improved organization, clearer presentation of statistics, and a stronger focus on gaps in the literature that the study aims to address.

1. Consider re-organizing paragraphs by topic sentence, followed by supporting statistics and sentences to improve flow and clarity.

2. Including 1-2 sentences outlining the gaps in existing literature this study is seeking to address can help frame the purpose and significance of the study.

3. It may be helpful to condense the statistical information to no more than a few statistics relevant to understanding the study’s context, to avoid overwhelming the reader.

Specifically, consider restructuring the content as below:

• Move the sentences starting: “Perinatal mortality remain….” and “An estimated 2.6 million babies are lost per year….” to the start of the introduction, followed by the definitions of perinatal mortality from the sentence starting: “Perinatal mortality refers to…” and ending with the sentence: “Whereas, a macerated stillbirth (MSB) shows…”

o Consider revising the sentence starting: “Whereas, a macerated stillbirth (MSB) shows….” by replacing “well before onset of labor” with “more than 8 hours prior to the onset of labor (35)” to provide clarity for the reader. Consider deleting the sentence in the discussion starting: “Macerated stillbirths are typically…” since that information is now in the introduction.

• Start a new paragraph with the sentence: “Efforts to reduce perinatal deaths…” The sentences starting at (p.3), “For instance, until recently, most stillbirths were excluded....” and ending at (p.4), “In stark contrast, SSA has remained with the highest…” could then be condensed and reorganized to: (recommendation of reviewer, but unsure if statistics are correct or that this is what the authors are trying to say)

o “Since the launch of the WHO Millennium Development Goals (MDG) in 2000 and the subsequent transition to the SDGs in 2015, global declines in annual PMR have been primarily attributed to improvements in HICs which account for only ~45% of worldwide cases.(16) While the WHO reported the global rate of perinatal mortality rate (PMR) in 2015 as ***[fill in rate]/1000 live births,(1) Hong Kong and the US have rates of 3.4 and 5.5/1000, respectively (17,18). This is in stark contrast with SSA, which has the highest regional PMR at 37.3/1000,(19) with Southern Africa alone having a PMR of approximately 30.3/1000 live births. These disparities demonstrate significant gaps between the PMRs in high-income countries and those in low- and middle-income countries (LMICs).

o Consider following these paragraphs with a new one on the statistics in Zambia by moving these sentences starting “Zambia is not immune to…” and “The country’s PMR stands at…” to the new paragraph.

o A new paragraph can read: “One contributing factor to the significant differences in PMR between HIC and LMIC is that, until recently, most stillbirths were excluded from global data tracking. The lack of social recognition, investment, and programmatic action has exacerbated the issue.(6) Additionally, 66% of worldwide maternal deaths occur in SSA, and recent studies demonstrate a strong link between maternal deaths and perinatal mortality, with an estimated 10 perinatal deaths occurring for every maternal death.”

Conclude this new paragraph with a gap in the literature, for example: “Despite this, evidence identifying predictors of PMR and effective interventions to reduce PMR in SSA and other LMICs remain limited.”

• Conclude the introduction with the study aims, as is currently written in the submitted manuscript.

• In the Methods section, “Eligible participants who were agreeable to participate…” sentence can be changed to “Participants were prospectively recruited as cases occurred, with matched controls, and informed written consent was obtained from all participants.”

Other comments:

Copy-editing of the manuscript is recommended to address minor spelling and grammatical errors. Some specific (but not comprehensive) examples include:

• In Abstract conclusions, take out the second comma, after “walking.”

• In the introduction, start a new paragraph for the sentence: “Perinatal mortality remains a significant GH challenge in low-income settings.” For the sentence after that, …”labor, and/or in the first seven days”

• In Methods, can take out eh second “first level hospitals” in the sentence starting with “The hospitals involved all the major hospitals…”

• In Methods, study design, sentence starting “In rare cases, …” spelling of comfortable

• In Methods, ethical considerations, 1st paragraph last sentence, “study’s purposed” = “study’s purpose” and “enrolment” = “enrollment”

• Only the first instance of “perinatal mortality rate” should be spelled out, the remainder should be abbreviated “PMR”

Overall, this paper provides valuable insights into the factors influencing perinatal mortality in low- and middle-income countries and has the potential to significantly contribute to the literature.

Reviewer #9: Dear Author,

Thank you for the good job, you need to revise my comments such as the adding the objectives and other needful issues highlighted in track changes.

Best regards

Reviewer #10: I would like to make a few suggestions that I think are important to the quality of the manuscript.

I also made comment on the paper attached for updated references, grammar errors.

1. The introduction includes a lot of data points that should be referenced to substantiate this intro.

For example - Source of 2.6 mill perinatal deaths - no reference given.

2. The Reference list is quite old for this section also. Particularly for the Maternal mortality estimates.

Please consider using the most recent estimates published in 2023 (https://www.who.int/publications/i/item/9789240068759) and note new estimates will be published by WHO in March 2025 - over the coming 2 weeks and ideally if you publish this month or early April - it would be good to use the March 2025 data which will give the paper an very current feel from the introduction.

Also, suggest to use the most recent newborn and stillbith estimate - UNIGME child estimates 2023 (disaggregated for newborn). https://childmortality.org/wp-content/uploads/2024/03/UNIGME-2023-Child-Mortality-Report.pdf and use UNIGME as a reference for stillbirth estimates - both globally and for Zambia:. https://childmortality.org/profiles/country?indicator=SBR

3. The statement the uses references 10-12 is not accurate.

The number of perinatal deaths for every maternal death is not evidence of a link.

The next sentence - that uses reference 13 is accurate. And is the key point. So the sentence for reference 10-12 can be deleted and the further substantial this statement made with reference 13 - which needs to be better substantiated:

I suggest Causes of Neonatal Mortality Liu et al 2016 or Reinebrandt 2018 which will helps to develop this point.

Its nicely summarised on slide 3 here whilc can also be referenced: https://www.healthynewbornnetwork.org/hnn-content/uploads/Blencowe_Stillbirths.pdf

4. Exclusion criteria - should there be a mention that participants are excluded if they do not give consent?

5. The risk factors set out a list of all distal and proximal factors included in the study - and for some there is no further analysis or discussion provided - for example congenital syphilis and congenital abnormalities, timelag to NICU?

It should be clarified that these are known factors for perinatal mortality but were not included in the study? And this should be clarified as a limitation of the study also?

Reviewer #11: The study addresses a critical public health issue by examining the predictors of perinatal mortality in Lusaka, Zambia. The manuscript is comprehensive and contributes valuable insights into maternal and neonatal health in a low-resource setting. However, a few minor issues need to be addressed.

First while the authors highlighted issues with data completeness and record-keeping, they should elaborate on how missing data were handled and what measures were taken to minimize information bias during data collection.

Second, the authors should enhance the explanation for the selection of specific predictor variables in the regression model.

In addition, the authors should provide a summary of policy implications or recommendations based on the findings to guide future interventions.

Reviewer #12: This manuscript addresses factors that are linked to perinatal mortality in the setting of Lusaka, Zambia. The methodology, data collection and analyses are sound. This manuscript has undergone several rounds of revisions and the authors have done an admirable job in responding to reviewers' comments. The paper is now more precise, though there are a few minor revisions that I would like to suggest to improve readability. With these changes noted below, I believe that the paper is ready for publication.

This latest revision still lacks line numbering, which makes it challenging to suggest specific revisions. I will use page number and attempt to identify the sentence where a revisions is suggested.

ABSTRACT: In the Results section, the second sentence in the "Results" section should be revised to "....compared TO early booking."

INTRODUCTION: On page 3 approximately 3/4 down the page rewrite as follows, "..and the lack of social recognition, investment, and programmatic action HAVE exacerbated the issue."

PAGE 6 near the top of the page, revise the sentence as follows, "...neonatal death, controls were FOLLOWED up before discharge......"

PAGE 17, near the top of the page, revise the sentence as follows, "with a primary education level were almost two times MORE likely to experience perinatal......"

PAGE 25 in the RECOMMENDATION section, I would suggest that you spell out "CEmONC" You provide the full name on page 21 (Comprehensive Emergency Obstetrics and Neonatal Care facilities) but some readers only read the Recommendations and it would be helpful to be clear as to what this abbreviation is.

**Do you want your identity to be public for this peer review?** For information about this choice, including consent withdrawal, please see our Privacy Policy

Reviewer #6: **Yes: ** Nsanzabera Charles

Reviewer #7: No

Reviewer #9: **Yes: ** Ebrima Bah

Reviewer #10: No

Reviewer #11: No

Reviewer #12: **Yes: ** Paul R De Lay, MD, DTM&H (Lond)

---

## [Decision Letter · Decision Letter 6]

21 May 2025

PGPH-D-24-01110R6

Predictors of perinatal mortality in the seven major hospitals of Lusaka Zambia: A Case Control Study

Dear Makasa,

Thank you for submitting your manuscript to PLOS Global Public Health. After careful consideration, we feel that it has merit but does not fully meet PLOS Global Public Health’s publication criteria as it currently stands. Therefore, we invite you to submit a revised version of the manuscript that addresses the points raised during the review process.

We look forward to receiving your revised manuscript.

Kind regards,

Collins Otieno Asweto, PhD

Academic Editor

Journal Requirements:

Reviewers' comments:

Reviewer's Responses to Questions

**Comments to the Author**

Reviewer #2: (No Response)

Reviewer #10: (No Response)

Reviewer #13: All comments have been addressed

Reviewer #14: (No Response)

publication criteria?

Reviewer #2: Partly

Reviewer #10: Yes

Reviewer #13: Yes

Reviewer #14: Yes

3. Has the statistical analysis been performed appropriately and rigorously?

Reviewer #2: No

Reviewer #10: Yes

Reviewer #13: Yes

Reviewer #14: Yes

4. Have the authors made all data underlying the findings in their manuscript fully available (please refer to the Data Availability Statement at the start of the manuscript PDF file)?

Reviewer #2: No

Reviewer #10: (No Response)

Reviewer #13: Yes

Reviewer #14: Yes

5. Is the manuscript presented in an intelligible fashion and written in standard English?

Reviewer #2: Yes

Reviewer #10: (No Response)

Reviewer #13: Yes

Reviewer #14: Yes

Reviewer #2: I have included several comments to the authors, within the PDF of the manuscript, attached.

Reviewer #10: Reviewer 10 comments.

My comments were not addressed for the opening section.

Currently the section reads with inaccurate information and data that is out of date which undermines this section greatly. This can be strengthened by correcting misinformation and using up to date data sources.

1. The background statement opens with positioning the issue using data that is 10 years old.

There are new estimates in 2025 for both newborn and stillbirths that should be used to position this paper with current needs.

2. The Sustainable Development Goals are not by WHO – The Sustainable Development Goals are by the United Nations. This needs to be corrected in the multiple places that 'WHO Sustainable Development Goals' are cited in the paper and the referencing needs to be accurate also.

3. Please use the WHO classification for stillbirth : which is a baby who dies after 28 weeks of pregnancy but before or during birth. The current reference is not valid. https://www.who.int/health-topics/stillbirth

4. Please include the up to date information that stillbirths are included as a target in the Every Newborn Action plan World Health Assembly Resolution (2014) https://apps.who.int/gb/ebwha/pdf_files/WHA67/A67_R10-en.pdf

and the UN has released stillbirth estimates since 2019. This information will support the point that stillbirth were invisible until very recently.

Reviewer #13: The manuscript is well-structured and presents important findings on predictors of perinatal mortality in Lusaka.

One minor observation is that while marital status data is presented—showing that nearly a quarter of participants were unmarried—the discussion does not explore its potential influence on care-seeking behavior or perinatal outcomes in the Zambian context. Including a brief comment on this could enhance the interpretive depth. Nevertheless, this omission does not detract from the overall quality of the study.

Reviewer #14: Most of the comments have been satisfactorily addressed by the author. However I noted that the comment in the methods section from a previous reviewer suggesting that socio-demographic variables be treated separately was not addressed. Additionally, the comment on handling missing data was not elaborate to explain it at the analysis stage.

Sample size calculation; First sentence should be corrected to “two controls per case”

Multivariate logistic model; the p-value for education was not significant, it should not be put in bold.

**Do you want your identity to be public for this peer review?** For information about this choice, including consent withdrawal, please see our Privacy Policy

Reviewer #2: **Yes: ** Andrew Kazibwe

Reviewer #10: **Yes: ** Olive Cocoman

Reviewer #13: No

Reviewer #14: **Yes: ** Sheillah Ansiima

---

## [Decision Letter · Decision Letter 7]

2 Aug 2025

Predictors of Perinatal Mortality in the Seven major Hospitals of Lusaka Zambia: A Case Control Study

PGPH-D-24-01110R7

Dear Makasa,

We are pleased to inform you that your manuscript 'Predictors of Perinatal Mortality in the Seven major Hospitals of Lusaka Zambia: A Case Control Study' has been provisionally accepted for publication in PLOS Global Public Health.

Best regards,

Collins Otieno Asweto, PhD

Academic Editor

Reviewer Comments (if any, and for reference):

Reviewer's Responses to Questions

**Comments to the Author**

Reviewer #13: All comments have been addressed

publication criteria?

Reviewer #13: Yes

3. Has the statistical analysis been performed appropriately and rigorously?

Reviewer #13: Yes

4. Have the authors made all data underlying the findings in their manuscript fully available (please refer to the Data Availability Statement at the start of the manuscript PDF file)?

Reviewer #13: Yes

5. Is the manuscript presented in an intelligible fashion and written in standard English?

Reviewer #13: Yes

Reviewer #13: The revised manuscript is well structured and is acceptable for publication in its present form

**Do you want your identity to be public for this peer review?** For information about this choice, including consent withdrawal, please see our Privacy Policy

Reviewer #13: No
